# MEMO: Memory-Augmented Model Context Optimization for Robust Multi-Turn Multi-Agent LLM Games

**Yunfei Xie** [* 1] **Kevin Wang** [* ‡ 2] **Bobby Cheng** [* 3] **Jianzhu Yao** [4] **Zhizhou Sha** [2] **Alexander Duffy** [5] **Yihan Xi** [2] **Hongyuan Mei** [6] **Cheston Tan** [3] **Chen Wei** [† 1] **Pramod Viswanath** [† 4] **Zhangyang Wang** [† 2]

## Abstract

Multi-turn, multi-agent LLM game evaluations exhibit substantial run-to-run variance. In long-horizon interactions, small deviations compound across turns and are amplified by multi-agent coupling, biasing win rate estimates and destabilizing comparative rankings across repeated tournaments. Prompt choice exacerbates this by inducing different effective policies and interaction dynamics. We address both instability and underperformance in games with **MEMO** (**Me**mory-augmented **MO**del context optimization), a self-play framework that treats inference-time context as an optimizable, agentic object by coupling **retention** and **exploration**. Retention distills self-play trajectories into a persistent memory bank of structured insights that act as priors; exploration combines tournament-style prompt evolution under TRUESKILL with prioritized replay over decisive states. Across five text-based games and a 2,000-game budget per task, MEMO raises mean win rate from 24.9% to 49.5% for GPT-4o-mini and from 20.9% to 45.0% for Qwen-2.5-7B, while sharply reducing run-to-run dispersion. These results suggest substantial headroom in multi-agent LLM game performance and robustness can be unlocked through memory, with MEMO achieving gains in negotiation and imperfect-information settings, while RL remains more effective in perfect-information settings. Project website: https://yunfeixie233.github.io/MEMO/.

[*]Equal contribution  [‡]Project Leader.  [†]Equal Advising.
[1]Rice University, Houston, TX, USA [2]The University of Texas at Austin, Austin, TX, USA [3]A*STAR, Singapore [4]Princeton University, Princeton, NJ, USA [5]Good Start Labs, USA [6]Toyota Technological Institute at Chicago, Chicago, IL, USA. Correspondence to: Yunfei Xie <xieyunfei01@gmail.com>, Kevin Wang <kevinwang.1839@gmail.com>, Bobby Cheng <bobbycxy1994@gmail.com>.

*Proceedings of the 43rd International Conference on Machine Learning*, Seoul, South Korea. PMLR 306, 2026. Copyright 2026 by the author(s).

## 1. Introduction

Large language models (LLMs) have rapidly saturated many static benchmarks, leaving limited headroom for single-turn QA and reasoning datasets such as AIME (AI-MO, 2024), SWE-Bench (Jimenez et al., 2024), and GPQA (Rein et al., 2024). This shifts attention toward multi-turn and interactive evaluations, namely game-based benchmarks (Duan et al., 2024; Topsakal et al., 2024; Fan et al., 2024), which stress long-horizon reasoning, adaptation, and strategic interaction. Games are easy to simulate, come with objectives, and require capabilities that apply to real-world challenges such as planning under uncertainty, negotiation, and context-sensitive decision making.

However, *multi-turn, multi-agent LLM evaluation is inherently unstable*. Because each model output becomes part of the subsequent input, small early deviations can compound across turns, leading to divergent trajectories (Laban et al., 2025). In multi-agent games, interaction coupling may exacerbate this effect: an inconsistent response from one agent can perturb the other agent's best responses, reshaping the joint trajectory (Cemri et al., 2025). Separately, some LLMs exhibit nondeterministic outputs even under nominally deterministic decoding settings (Blair-Stanek & Van Durme, 2025). From an evaluation perspective, these factors can bias win-rate estimates and destabilize comparative rankings across repeated tournaments, complicating reproducibility and fair model comparison.

Inference-time *context*, including prompts, instructions, and auxiliary information, offers a direct lever for both performance and robustness in interactive settings. Context shapes how observations are interpreted, which actions are considered valid, and how decision rules apply across turns. As a result, small contextual variations can induce different effective policies and interaction dynamics, leading to large performance shifts and even rank reversals across models (see Appx. A). These observations motivate treating inference-time context not as a fixed wrapper, but as an *agentic object* that should be optimized and evaluated under interaction.

Existing approaches, however, struggle in multi-turn, path-dependent games. Prompt engineering techniques such as

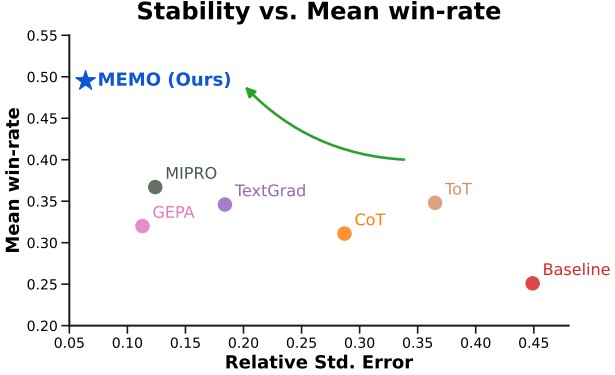 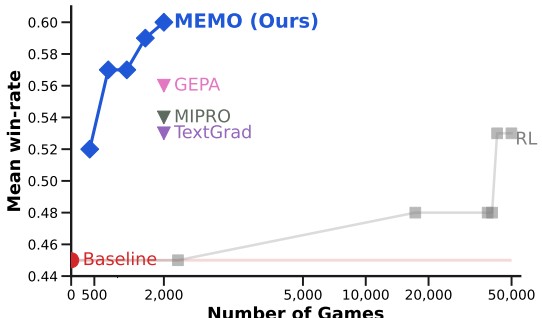

*(a)* **Performance and stability across different methods.** *(b)* **Training Efficiencies in KUHNPOKER.**

**Figure 1.** **Left:** Run-to-run performance and stability comparison: Using GPT-4o-mini with MEMO achieves the highest mean win rate (49.5%) with the lowest RSE (6.4%). **Right:** Learning efficiency comparison against the self-play RL baseline method UNSTABLEBASELINE: Using Qwen2.5-7B-Instruct with MEMO reaches 60% win rate on Kuhn Poker with 2,000 games.

chain-of-thought (CoT) (Wei et al., 2022) instructions or hand-designed templates remain fixed throughout evaluation. While these can improve win rate or reduce superficial errors, they do not adapt to failure modes or strategic patterns that emerge through interaction. Automatic prompt optimization methods (Yuksekgonul et al., 2025; Yin & Wang, 2025; Agrawal et al., 2025; Opsahl-Ong et al., 2024) allow prompts to adapt, but are largely developed for static tasks. They update prompts using feedback from a local batch of trajectories and lack persistent memory. In multi-turn, multi-agent games, different tournaments surface different decisive states and rare failure modes; without a mechanism to retain and reuse insights across rounds, prompt optimization becomes run-dependent, leading to high variance in both learned contexts and performance.

We therefore propose **MEMO** (**Me**mory-augmented **MO**del context optimization), a weight-free self-play framework built on two coupled components: *(i) Retention* and *(ii) Exploration*. The *exploration* component of MEMO searches over candidate contexts to discover effective policies. The *retention* component preserves and combines insights across optimization generations. Together, these components treat inference-time context as an optimizable policy object and stabilize optimization through shared memory.

We evaluate performance and stability on five text-based games from `TextArena` and `SPIN-Bench` (Guertler et al., 2025a; Yao et al., 2025). MEMO achieves large, budget-efficient gains and more reliable evaluation. Mean win rate improves from **24.9%** to **49.5%** for **GPT-4o-mini** (OpenAI, 2024a) and from **20.9%** to **45.0%** for **Qwen-2.5-7B-Instruct** (Yang et al., 2024), using only 2,000 self-play games per task. Moreover, treating evaluation as a context-constructive process yields rankings that are more consistent across runs and small prompt variations.

Our contributions are threefold:

- **Context sensitivity in multi-turn, multi-agent LLM games.** We show that evaluation outcomes depend on contextual choices. Small prompt variations can induce shifts in effective policies and comparative rankings, motivating robust checks such as prompt-stratified reporting rather than reliance on single-prompt evaluations.

- **MEMO as a plug-in recipe for multi-agent games.** We introduce a framework that unifies structured reflection, persistent memory, context evolution, and prioritized replay into a single recipe for multi-agent settings, addressing shortfalls of batch-local context updates. The contribution is the cross-generation coupling of memory and exploration. Ablating either component substantially reduces performance, while combining them yields gains larger than the sum of each alone, as shown in Tab. 3.

- **Training-efficiency gains with improved stability.** We report that MEMO substantially improves win rates under a fixed self-play budget while reducing run-to-run variance of end-to-end outcomes. It achieves competitive or stronger results than existing prompt optimization methods in imperfect information games, while RL remains more effective in perfect-information settings.

**Conflict of Interest Disclosure.** This work received financial support from Good Start Labs and Sentient to cover experiment costs. Author Alexander Duffy is affiliated with Good Start Labs. No proprietary models, datasets, or systems developed by Good Start Labs or Sentient are evaluated in this paper; all experiments use publicly available LLMs (GPT-4o-mini, Qwen-2.5-7B-Instruct, Gemini-2.5-Flash-Lite, Grok-4-Fast-Non-Reasoning, Qwen3-235B-A22B-Instruct-2507) and open environments (TextArena, SPIN-Bench, MultiAgentBench). The other authors declare no financial conflicts of interest related to this work.

## 2. Preliminary and Problem Statement

**Two-Player Multi-Turn Markov Game.** We formalize the setting as a two-player, turn-based, zero-sum, partially observable Markov game specified by the tuple $(S, A, O, P, \Omega, \rho)$. Here $S$ is the state space of game configurations, $A$ is the action space where each action is a complete model response (i.e., a full turn rather than a single token), and $O$ is the observation space. The transition kernel $P : S \times A \to \Delta(S)$ governs state dynamics, while the observation function $\Omega : S \to O$ maps states to partial observations (e.g., a player's private cards in poker). The outcome function $\rho : S_{\text{term}} \to \{-1, 0, 1\}$ assigns loss, draw, or win to Player 0 at terminal states $S_{\text{term}} \subset S$.

Players alternate turns. At step $t$, the active player $p_t = t \bmod 2$ observes $o_t = \Omega(s_t)$, selects an action $a_t \in A$, and induces a transition $s_{t+1} \sim P(s_t, a_t)$. The trajectory $\tau = (s_0, a_0, \ldots, s_H)$ terminates when a terminal state is reached in $H$ steps (game length), yielding outcome $r_0(\tau) = \rho(s_H)$ for Player 0 and $r_1(\tau) = -r_0(\tau)$.

**Game Context: Prompt and Memory.** We define *context* as all information that conditions the model before and during play. Let $c = (q, M)$, where $q$ is the instruction prompt, including role and system text fixed at game start, and $M$ is the memory injected at inference time without weight updates. $M$ consists of structured, reusable insights distilled from past self-play trajectories. In MEMO, $M$ is drawn from a persistent memory bank $\mathcal{B}_{\text{mem}}$ that accumulates across optimization iterations, and each game instance may use a subsampled memory $M \subseteq \mathcal{B}_{\text{mem}}$.

**Full-Context Evaluation.** We evaluate each method over $n$ independent runs of its full context-optimization pipeline, each producing a final context $c^*$ that is evaluated on a fixed game suite $\mathcal{G}$. For each game, we play a fixed number of rounds against a fixed opponent pool, swapping first-move order to reduce bias (opponents use the reference contexts in Appx. F). Let $x_r$ denote the run-level performance, defined as the mean win rate averaged over all games, opponents, and rounds. We report the mean performance across runs, $\text{mean}(x_1, \ldots, x_n)$, together with the relative standard error $\text{RSE}(\%) = 100 \times \frac{\text{std}(x_1, \ldots, x_n)}{\text{mean}(x_1, \ldots, x_n)\sqrt{n}}$, where lower RSE indicates greater run-to-run stability.

## 3. The MEMO Framework

We present **MEMO**, a weight-free self-play framework that optimizes inference-time context in two-player Markov games. A key goal is to make context optimization *less run-dependent*. Standard optimizers update prompts from the current batch only, so insights discovered in one round are lost in the next; this causes independent runs to drift toward different solutions. MEMO addresses this by aug-

menting context optimization with a persistent *memory bank* (Sec. 3.2) that accumulates reusable priors across optimization generations, allowing later rounds to build on lessons from earlier ones rather than rediscovering them.

MEMO operates over multiple optimization generations. Each generation $g$ consists of a self-play tournament, context evolution (Sec. 3.1), insight extraction from trajectories (Sec. 3.2), and state selection for replay (Sec. 3.3). Fig. 2 provides an overview, Subsec. 4.2 lists our hyperparameter selection, and Appx. B discusses our hyperparameter tuning.

### 3.1. Tournament-Based Context Optimization

This section describes the *exploration* component of MEMO, which searches over candidate contexts to discover effective policies. At a high level, MEMO maintains a population of $N$ candidate contexts, each defining a different prompt and set of priors for the agent. These candidates are evaluated via multi-agent self-play: each agent, equipped with its candidate context, plays a tournament of games against a fixed baseline agent (the same base model using only a default prompt). From these games we observe whether each context leads to wins or losses. The basic selection rule is simple: keep contexts that win reliably, discard those that lose, and generate new candidate contexts to fill the population for the next generation. This loop repeats across optimization generations, progressively searching for more effective contexts.

**Context Selection via Game Outcomes.** The core idea is to evaluate each candidate context by its game performance: contexts that lead to wins are retained for the next generation, while those that lead to losses are discarded. Let $\mathcal{C}_g$ denote the *context population* at optimization generation $g$. Each context $c \in \mathcal{C}_g$ is evaluated via multi-agent self-play in games against a baseline agent (see Appx. F). For asymmetric games, each round consists of two games with roles swapped to remove first-move bias. These matches produce win/loss outcomes for each context, but raw win counts are unreliable when games are limited: a context that wins 3 out of 3 games may simply be lucky rather than genuinely strong. To address this, we use TRUESKILL (Herbrich et al., 2006), a Bayesian skill rating that models each context's skill as a Gaussian with mean $\mu_c$ and uncertainty $\sigma_c$. We select contexts using a conservative lower-confidence bound:

$$S(c) = \mu_c - \kappa \, \sigma_c, \tag{1}$$

where $\kappa$ is a penalty coefficient (see Sec. 4.2). This penalizes contexts with high uncertainty, favoring those that win reliably across multiple observations.

**Context Generation for the Next Generation.** After selection, low-scoring contexts are discarded, leaving the pop-

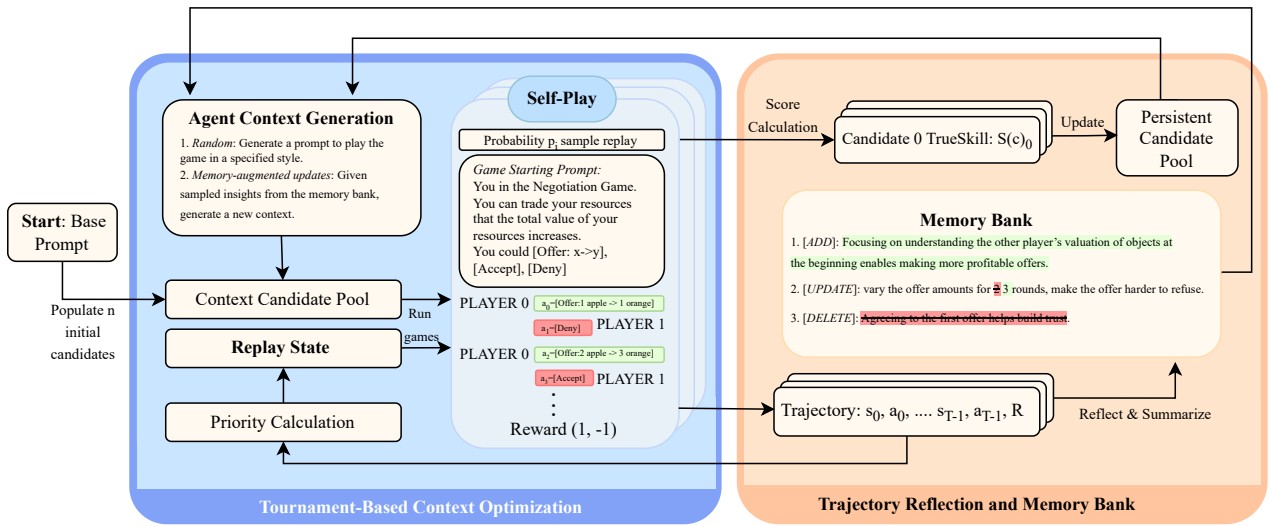

*Figure 2.* **The MEMO Framework**. At each optimization generation, new candidate contexts are proposed through two strategies: random proposals and memory-augmented updates. These candidates are then evaluated via self-play, and the best-performing candidates are used to update the pool for the next generation. To encourage exploration and mitigate redundant early moves, a prioritized replay module is introduced, enabling efficient search for robust prompts and priors within a single game.

ulation incomplete. To restore the population to size $N$ for the next generation, we generate new candidate contexts. Across optimization generations, we maintain a *persistent candidate pool* $\mathcal{P}$ that stores the best contexts observed so far. After evaluating the current population $\mathcal{C}_g$, we update $\mathcal{P}$ by retaining only the top-scoring candidates from $\mathcal{P} \cup \mathcal{C}_g$. We then form the next generation's population $\mathcal{C}_{g+1}$ using two proposal operators, where a fraction of new candidates are generated via random proposals and the remainder via memory-augmented updates (see Sec. 4.2 for the specific ratio):

1. **Random proposals**: introduce novel variations to encourage exploration by sampling a playstyle from a fixed catalog and applying small, length-bounded edits to the base context to instantiate that style while preserving legality and interface constraints (Appx. C.1).

2. **Memory-augmented updates**: incorporate insights extracted from trajectory reflections (Sec. 3.2) into targeted prompt edits.

Note that in the first generation ($g = 0$), the memory bank is empty, so all initial contexts are generated via random proposals.

After the final optimization generation, MEMO outputs the highest-scoring context in $\mathcal{P}$:

$$C^\star = \arg\max_{C \in \mathcal{P}} S(C).$$

### 3.2. Trajectory Reflection and Memory Bank

This section describes the *retention* component of MEMO, which preserves and combines insights across optimization

generations. Multi-turn games make post-hoc attribution easier than online decision making because a completed trajectory reveals which choices led to the observed outcome, relating to hindsight-style analysis (Andrychowicz et al., 2017). MEMO exploits this by extracting structured insights from completed self-play trajectories and storing them in a persistent memory bank.

**Trajectory Reflection.** After each optimization generation, we sample a fixed number of completed self-play trajectories and prompt the model to extract a small set of typed insights, such as rule clarifications, legality constraints, and strategy priors. For each sampled trajectory, the model reviews the sequence of states, actions, and final outcome, then produces one or more candidate insights that summarize lessons learned. These insights capture what worked, what failed, and why, providing structured feedback that can inform future play. The reflection prompt template is provided in Appx. D.

**Memory Bank.** MEMO maintains a shared memory bank $\mathcal{B}_{\text{mem}}$ that persists across optimization generations. For each generation with $N$ evaluated trajectories, the reflection step produces up to $N$ candidate insights that must be reconciled with the existing memory bank. Following database-style operations (Martin, 1983), we merge new insights into $\mathcal{B}_{\text{mem}}$ using three operations:

1. **Add**: If a new insight is not similar to any existing insight in the memory bank, it is added directly.

2. **Remove**: If a new insight conflicts with an existing insight (i.e., they suggest contradictory strategies or

conclusions), both the new and existing insights are removed to avoid misleading the agent.

3. **Edit**: If a new insight is similar to an existing one, the two are merged by enhancing, generalizing, or improving the existing insight to be more actionable.

The agent compares each candidate insight against the current memory bank and applies the appropriate operation. This merge procedure allows the memory bank to grow, refine, and self-correct over time. The memory operation prompt is provided in Appx. E.

In the next optimization generation, we sample a compact subset $M \subseteq \mathcal{B}_{\text{mem}}$ and append it to the context of a fraction $\pi$ of the candidate population during self-play (i.e., $\pi$ controls what proportion of agents receive memory-based initialization), providing reusable, game-specific priors at inference time (see Sec. 4.2 for specific values). The same memory bank also conditions the memory-augmented proposal operator, enabling targeted prompt edits that reuse aggregated lessons rather than relying only on the most recent tournament.

### 3.3. Prioritized Replay

Trajectory reflection improves retention, but exploration alone does not guarantee that rare or decisive states will be revisited. To improve trajectory coverage, MEMO maintains a replay buffer $\mathcal{B}_{\text{rep}}$ that stores trajectory prefixes together with the environment seed needed to reproduce them. Because storage occurs at each turn within an episode, replayed trajectories need not cover a full game. Invalid moves are retained to preserve the *unaltered course of play*, ensuring that replays faithfully reflect the original gameplay dynamics. To avoid dominance by common action patterns, the buffer *biases sampling toward infrequently encountered trajectories*, encouraging a more diverse and balanced pool of prompt-level insights. We prioritize rare prefixes using an inverse-frequency score, defined for a stored prefix $\tau$ as priority$(\tau) = \frac{1}{\text{count}(\tau)}$. During sampling, the probability $p_i$ of selecting trajectory $\tau_i$ is obtained by raising its priority to a power $\alpha > 0$ and normalizing over the buffer, $p_i = \frac{\text{priority}(\tau_i)^\alpha}{\sum_{j=1}^{|\mathcal{B}_{\text{rep}}|} \text{priority}(\tau_j)^\alpha}$, where $|\mathcal{B}_{\text{rep}}|$ denotes the current number of stored trajectories.

The buffer is first populated during generation 0 and becomes available from generation 1 onward. A gating parameter $\beta$ (*replay probability*) determines how often games are initialized from the replay buffer rather than played afresh. When replay is chosen, the stored trajectory prefix (i.e., the sequence of past player actions, corresponding game states and the associated game's random seed) are injected into the environment, ensuring faithful reproductions of past episodes while balancing new exploration. Specific values for $\alpha$, $\beta$, and buffer capacity $B$ are provided in Sec. 4.2.

## 4. Experiment Setup

### 4.1. Game Environments

Following prior interactive evaluation suites such as LMGame-Bench and BALROG (Hu et al., 2025; Paglieri et al., 2025), our games span core problem classes studied in game theory and multi-agent systems. They are categorized as: **Negotiation** games, which test cooperation, compromise and strategic trade-offs (Kramár et al., 2022; Abdelnabi et al., 2024b); **Imperfect Information** games, which require reasoning under uncertainty and partial observability (Brown et al., 2020; Guo et al., 2024); and **Perfect Information** games, which emphasize planning and long-horizon decision-making with full state visibility (Silver et al., 2017a). See Appx. J for environment descriptions.

### 4.2. Hyperparameter Selection

Following our methodology as discussed in Sec. 3, we use a fixed set of hyperparameters throughout all experiments. The key design choices in MEMO are as follows:

**Context optimization loop:** We maintain a candidate population of size $N = 8$ (i.e., 8 candidate contexts evaluated in parallel) over 5 optimization generations, with 50 self-play games per candidate per optimization generation. The TrueSkill penalty coefficient is set to $\kappa = 1$.

**Memory-augmented initialization:** We control what proportion of the candidate population receives insights from the shared memory bank $\mathcal{B}_{\text{mem}}$ at initialization. We denote this proportion by $\pi \in [0, 1]$, where $\pi = 0$ means no candidates receive memory and $\pi = 1$ means all candidates are initialized with sampled insights. We use $\pi = 0.75$.

**Replay mechanism:** The replay mechanism introduces three hyperparameters: the buffer capacity $B$, which specifies the maximum number of stored trajectories; the priority exponent $\alpha$, which controls how strongly rare trajectories are prioritized; and the replay gate $\beta$, which determines the probability of initializing from replay rather than starting a fresh game. We use $B = 100{,}000$, $\alpha = 0.6$, and $\beta = 0.4$.

More details of our ablation are covered in Appx. B.

### 4.3. Optimizer Settings

**Baseline:** Our baseline uses the default `TextArena` (Guertler et al., 2025a) prompts without optimization. See Appx. F for examples.

**MEMO:** Using the MEMO Framework with hyperparameters specified in Sec. 4.2, each self-play tournament corresponds to one optimization generation. Reflection signals are incorporated into the optimization, and token costs of each method are reported in Tab. 2.

*Table 1.* Benchmark results for different approaches using GPT-4o-mini and Qwen2.5-7B-Instruct across multiple tasks. Each win rate is the mean across three evaluation models. **Type** denotes the optimization paradigm: Static prompting, Prompt optimization, Reinforcement learning (RL), and our method. For full model-wise results, see Appendix H.

| Type | Optimizer | Negotiation | | Imperfect Info | | Perfect Info | Mean Win Rate | Mean RSE |
| | | SimpleNegotiation | TwoDollar | KuhnPoker | Briscola | SimpleTak | | |
|---|---|---|---|---|---|---|---|---|
| **GPT-4o-mini** | | | | | | | | |
| STATIC | **baseline** | 31.3% | 32.2% | 39.1% | 0.3% | 21.4% | 24.9% | 31.1% |
| | CoT | 27.8% | 25.7% | 46.5% | 30.4% | 24.8% | 31.1% | 28.7% |
| | ToT | 26.3% | 27.0% | 51.7% | 45.1% | 23.8% | 34.8% | 36.5% |
| PROMPT | TextGrad | 42.0% | 44.6% | 55.6% | 7.1% | 23.6% | 34.6% | 18.4% |
| | MIPRO | 38.4% | 50.9% | 55.1% | 19.7% | 19.1% | 36.7% | 12.4% |
| | GEPA | 36.8% | 40.4% | 52.2% | 3.3% | 26.9% | 32.0% | 11.3% |
| OURS | **MEMO** | **54.9%** | **52.4%** | **55.6%** | 42.7% | **41.8%** | **49.5%** | **6.4%** |
| **Qwen2.5-7B-Instruct** | | | | | | | | |
| STATIC | **baseline** | 24.0% | 17.1% | 45.3% | 2.8% | 15.1% | 20.9% | 20.9% |
| | CoT | 23.8% | 18.7% | 42.0% | 25.8% | 13.6% | 24.8% | 43.4% |
| | ToT | 27.1% | 20.7% | 42.2% | 22.7% | 15.1% | 25.6% | 40.2% |
| PROMPT | TextGrad | 37.1% | 29.3% | 52.8% | 7.1% | 22.4% | 29.7% | 21.7% |
| | MIPRO | 42.4% | 47.5% | 53.8% | 2.2% | 20.9% | 33.4% | 7.3% |
| | GEPA | 34.4% | 31.7% | 55.8% | 2.7% | 19.3% | 28.8% | 14.8% |
| RL | UnstableBaseline | 41.1% | 30.4% | 52.7% | **53.3%** | 47.3% | **45.0%** | 43.3% |
| | SPIRAL | 45.7% | – | 56.7% | – | 32.7% | – | – |
| OURS | **MEMO** | **48.0%** | **48.4%** | **60.0%** | 31.5% | 36.9% | **45.0%** | **7.0%** |

For comparison, we benchmark against `Textgrad` (Yuksekgonul et al., 2025), `MIPRO` (Opsahl-Ong et al., 2024), and `GEPA` (Agrawal et al., 2025), as well as RL baselines including `UnstableBaseline` (Guertler et al., 2025b) and `SPIRAL` (Liu et al., 2025). Detailed setups of their optimization are provided in Appx. G.

### 4.4. Evaluation Settings

All experiments use **GPT-4o-mini** (OpenAI, 2024a) and **Qwen-2.5-7B-Instruct** (Yang et al., 2024) as base models. For prompt-based methods, we perform **three** independent runs. In each run, the optimized prompt and context are evaluated against held-out opponents: Grok-4-Fast-Non-Reasoning (xAI, 2025a), Gemini-2.5-Flash-Lite (Comanici et al., 2025), and Qwen3-235B-A22B-Instruct-2507 (Yang et al., 2025). Unless otherwise noted, each run consists of 50 games. We report mean win rates across runs together with relative standard error (RSE). A fixed sampling temperature of $\tau = 1.0$ is used throughout, matching the default setting used by many LLM APIs. We further verify robustness to this choice with a KuhnPoker temperature sweep over $\tau \in \{0.5, 0.75, 1.0\}$, where MEMO remains stronger than both baselines and evaluation variance does not vary monotonically with temperature (Appendix G.1).

For RL-based methods, we train a single policy, select the best checkpoint, and evaluate it over **three** sets of 50 games each against the same opponents. The mean win rate and RSE across these sets are reported in Tab. 1.

## 5. Results and Analysis

**Observation 1: Persistent self-play memory enables stable, sample-efficient gains.** As shown in Tab. 1, MEMO consistently outperforms other prompt optimization methods, achieving an average gain over `TextGrad` (14.9%), `MIPRO` (12.8%), and `GEPA` (17.5%) with GPT-4o-mini. While the margin relative to RL-based methods such as `UnstableBaselines` and `SPIRAL` is smaller, MEMO remains competitive while using **19× fewer** environment interactions (**2,000** vs. **38,000** games).

These results are consistent with MEMO's ability to reuse information across episodes in the form of persistent, game-specific insights. Qualitative analysis of the memory bank (Appx. K) shows that high-quality entries capture transferable strategic principles, such as time pressure in negotiation, asymmetric resource valuation, or pressure-based betting, rather than instance-specific action reminders. This allows useful abstractions to persist across optimization generations, while less informative or overly specific feedback is gradually diluted. Unlike prompt-only optimization methods that reset context after each update, MEMO retains information across optimization generations, allowing performance improvements to accumulate with substantially fewer interactions. We investigate the contribution of this mechanism in Observation 2.

Cross-episode information reuse reduces variance in multi-turn gameplay. The 'baseline' runs in Tab. 1 exhibit high variances, likely due to the compounding effects of early decision errors. While other prompt optimization methods were able to reduce the RSE (defined in Sec. 2)

*Table 2.* Output token cost for each prompt optimization method (exact counts).

| Optimizer | SimpleNegotiation | KuhnPoker | SimpleTak | Avg. Tokens |
|---|---|---|---|---|
| Textgrad | 842 | 986 | 938 | 922 |
| MIPRO | 145,864 | 162,084 | 754,534 | 354,161 |
| GEPA | 110,325 | 119,365 | 111,907 | 113,865 |
| **MEMO (Ours)** | 87,364 | 94,160 | 89,152 | 90,225 |

*Table 3.* GPT-4o-mini ablation experiments with progressive module additions. Rows shaded indicate use of **Memory**.

| Modules | | | Win Rate | | | Summary | |
|---|---|---|---|---|---|---|---|
| Tournament | Mem | Replay | TwoDollar | KuhnPoker | Briscola | Mean | $\Delta_{base}$ |
| | | | 32.2% | 39.1% | 0.3% | 23.8% | – |
| ✓ | | | 24.7% | 54.7% | 2.0% | 27.1% | +3.3 |
| | ✓ | | 34.2% | 42.0% | 26.3% | 34.2% | +10.4 |
| ✓ | | ✓ | 32.0% | 54.2% | 38.7% | 41.6% | +17.8 |
| ✓ | ✓ | | 48.7% | 57.2% | 38.4% | 48.1% | **+24.3** |
| ✓ | ✓ | ✓ | 52.4% | 55.6% | 42.7% | 50.2% | **+26.4** |

when compared to the 'baseline', it was MEMO that consistently achieved the lowest mean RSE across games; for example, on GPT-4o-mini, MEMO attains an average RSE of 6.4%, compared to MIPRO's 12.4%. Notably, UnstableBaselines showed increased RSE, indicating that outcome-based RL with sparse end-game rewards remains unstable in multi-turn, multi-agent settings (Wang et al., 2025). Together, these results demonstrate that cross-episode information reuse is crucial for both performance and stability.

Retaining high-value insights also improves computational efficiency. As shown in Tab. 2, MEMO uses only 91K output tokens on average, about one-quarter of MIPRO (354K) and 20% fewer than GEPA (113K), while achieving similar or better win rates (Tab. 1). In contrast, methods such as MIPRO and GEPA rely on many reflective rollouts and prompt revisions, increasing token usage without commensurate performance gains, while TextGrad uses very few tokens (~1K) but lacks capacity to learn complex multi-turn behaviors. By retaining high-value insights and reusing them across generations, MEMO concentrates learning on fewer, more informative interactions, improving the trade-off between token cost, interaction budget, and win rate.

**Observation 2: Retention and structured exploration are both necessary for LLM learning in multi-turn game settings.** We observe that retention and structured exploration are both important components in MEMO based on the ablations in Tab. 3. All variants use prompt optimization and differ only in whether they maintain a persistent **Memory** bank as retention and whether they enrich trajectories via tournament play and replay as exploration. In the tournament-only setting, prompt updates are computed from the current tournament trajectories and no insights are stored across generations.

The ablation ladder indicates that a memory bank yields large gains when we use multi-generation tournament exploration. Mean win rate increases from 23.8% with prompt optimization to 27.1% with tournament-only (+3.3) and to 34.2% with Memory-only (+10.4), then to 41.6% with tournament plus replay (+17.8). The largest jump occurs when tournament exploration is paired with Memory, reaching 48.1% (+24.3), and replay adds a smaller improvement to 50.2% (+26.4). Prior work shows that random exploration paired with learning can produce substantial gains in multi-turn settings (Chen et al., 2025). Our results refine this picture for persistent memory. Random exploration is not enough to reliably populate a memory bank with transferable, high-signal insights, and structured exploration provides the repeated evaluation needed to filter what gets retained. This view matches population-based game learning, where robustness is driven by repeated evaluation against diverse opponents and mixtures rather than unstructured exploration (Lanctot et al., 2017).

**Observation 3: Learned contexts can generalize across games.** In realistic multi-task settings, agents rarely operate within a single environment in isolation. Hence, an effective retention mechanism should capture transferable interaction priors that generalize across games, enabling reuse of learned context without task-specific fine-tuning. Tab. 4 reports our cross-game results.

**Protocol-level skill transfer across game families.**

Core interaction components such as turn management, action formatting, and short-horizon planning can generalize even when payoff structures differ. For example, transferring context from SIMPLETAK → KUHNPOKER improves performance by +25.9%, and TWODOLLAR → SIMPLETAK yields a +26.4% gain. This retained context can act as a general decision scaffold beyond game-specific heuristics. We provide a case study in Appx. L.

**Transfer exhibits directional asymmetry.**

Transfer effectiveness depends on the direction of knowledge transfer. While context from SIMPLENEGOTIATION improves performance on TWODOLLAR (+5.6%), the reverse shows negligible effect (−0.2%). Similarly, BRISCOLA → SIMPLETAK shows negative transfer (−7.1%). This asymmetry indicates that successful transfer depends on structural alignment between source and target games.

**Observation 4: Learned context does not always transfer across models.** As shown in Fig. 3, we test whether a context learned via self-play with GPT-4o-mini can generalize across models. Specifically, we apply the learned prompts and retained experience produced by MEMO for GPT-4o-mini to Gemini-2.5-Flash-Lite and Grok-4-fast-non-reasoning and evaluated them against the same opponent pool described in Sec. 4.4. We observe that the context

*Table 4.* Generalization across tasks. We read each cell as a transfer *source → target*: rows (↓) name the source game where MEMO learns context through self-play, and columns (→) name the target game where the same context is evaluated *zero-shot* without further training. Diagonal entries are the in-domain case. Each cell reports the target win rate averaged over 50 independent matches, with the change relative to the target's no-context baseline (Tab. 1) in parentheses.

| ↓ Source Game | Target Game (zero-shot evaluation) ⟶ | | | | | Mean |
|---|---|---|---|---|---|---|
| | Negotiation | | Imperfect Info | | Perfect Info | |
| *(self-play training)* | SimpleNegotiation | TwoDollar | KuhnPoker | Briscola | Simpletak | Win Rate |
| **GPT-4o-mini** | | | | | | |
| SimpleNegotiation | 46.9% (+15.6%) | 37.8% (+5.6%) | 48.9% (+9.8%) | 0.0% (-0.3%) | 37.7% (+16.3%) | 34.3% (+9.4%) |
| TwoDollar | 31.1% (-0.2%) | 48.7% (+16.5%) | 53.3% (+14.2%) | 1.1% (+0.8%) | 47.8% (+26.4%) | 36.4% (+11.5%) |
| KuhnPoker | 31.1% (-0.2%) | 34.4% (+2.2%) | 57.2% (+18.1%) | 22.2% (+21.9%) | 30.0% (+8.6%) | 35.0% (+10.1%) |
| Briscola | 38.9% (+7.6%) | 27.8% (-4.4%) | 57.8% (+18.7%) | 38.4% (+38.1%) | 14.3% (-7.1%) | 35.4% (+10.6%) |
| Simpletak | 37.8% (+6.5%) | 35.6% (+3.4%) | 65.0% (+25.9%) | 0.0% (-0.3%) | 30.7% (+9.3%) | 33.8% (+9.0%) |

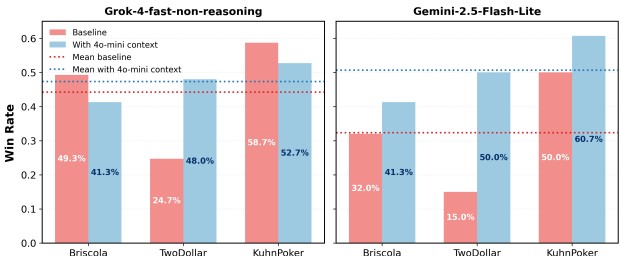

*Figure 3.* Performance of models across games with and without GPT-4o-mini context.

learned transfers effectively to Gemini-2.5-Flash-Lite, yielding consistent improvements across the three environments. Namely, the gains are most pronounced in TWODOLLAR, where Gemini exhibits a substantial increase in win rate when augmented with the 4o-mini context.

In contrast, the same context does not yield consistent benefits for Grok-4-fast-non-reasoning, as seen in the performance drops in BRISCOLA and KUHNPOKER. This suggests that learned contexts capture reusable strategic structures that can generalize across certain model families, but its effectiveness depends on how the target model processes and acts on the contextual information for gameplay.

# 6. Related Work

## 6.1. Prompt Optimization

Automatic prompt optimization has evolved into a principled, black-box search over prompt seeds, feedback signals, candidate generation, and selection strategies (Ramnath et al., 2025). Programmatic frameworks such as `DSPy` compile LM pipelines and optimize prompts directly toward a user metric (Khattab et al., 2024); gradient-via-text methods propagate natural-language feedback through computation graphs to update intermediate decisions (Yuksekgonul et al., 2025). Recent systems jointly search over agentic patterns and prompt contents (Spiess et al., 2025), offer zero-

configuration prompt pipelines with meta-optimizers and `DSPy` backends (Murthy et al., 2025), or meta-learn general system prompts while adapting user prompts (Choi et al., 2025). MEMO complements this line by targeting interactive games: it evolves context via conservative selection, writes structured insights to a persistent memory bank, and reuses them across turns. It provides rule-aware priors without weight updates while remaining backbone-agnostic. For a detailed comparison of our approach and existing prompt optimization methods, please refer to Appx. H.

## 6.2. LLM for Games

Early multi-agent evaluations used role prompts and multi-turn dialogue to probe cooperation and theory-of-mind (Abdelnabi et al., 2024a). Community arenas expanded coverage: `TextArena` provides competitive text games with online TrueSkill ranking (Guertler et al., 2025a); `SPIN-Bench` combines planning, cooperative/competitive play, and negotiation, highlighting limits in deep reasoning and coordination (Yao et al., 2025); and `GT-Bench` evaluates strategic play in board and card games (Duan et al., 2024). Prompt design strongly affects move quality (Topsakal et al., 2024), and moving toward off-the-shelf games required harnesses to reduce perception and prompt brittleness (Hu et al., 2025). We provide an empirical analysis of prompt-induced ranking instability in Appx. A. MEMO addresses this brittleness in text-based game settings directly: it treats evaluation as agentic context construction, stabilizing rankings under prompt variation while improving adherence to game capabilities underexplored by fixed-prompt protocols.

## 6.3. Self-Play and Evolutionary LLMs

Classical self-play (`AlphaGo`/`AlphaZero`) established competitive self-improvement through repeated matches and selection (Silver et al., 2017b; 2016). LLM variants close the loop without large curated corpora: `Absolute Zero` leverages data-free RLVR to attain strong math/cod-

ing results (Zhao et al., 2025); SPIRAL frames multi-turn reasoning as zero-sum self-play (Liu et al., 2025); and language self-play improves instruction following via self-generated interactions (Kuba et al., 2025). Evolutionary approaches perform reflective prompt/program search (e.g., GEPA outperforming RL baselines; evolutionary coding agents) (Agrawal et al., 2025; Novikov et al., 2025). Separately, VOYAGER (Wang et al., 2023) builds a composable skill library of executable code for open-ended single-agent exploration, accumulating capabilities over time but without adversarial self-play or strategic memory consolidation. ExpeL (Zhao et al., 2024) extracts and accumulates natural-language insights from trajectories, but likewise operates in single-agent settings and modifies context only through insight injection and retrieval. STRATEGIST (Light et al., 2025) targets adversarial multi-agent games but optimizes executable strategy artifacts (value heuristics as code, dialogue guides as worksheets) refined by external MCTS rather than the inference-time context itself. MEMO combines and extends these ideas in a tuning-free way: it performs evolutionary context search guided by a reliability-aware objective (TrueSkill), augments it with persistent memory to supply game-specific priors, and uses prioritized replay to revisit rare informative states, yielding stronger and more reliable in-game performance without parameter updates or external search at play time.

## 7. Conclusion

We addressed run-to-run variance in multi-turn, multi-agent LLM evaluation caused by compounding deviations and prompt sensitivity. We introduced MEMO, a weight-free self-play framework that couples *retention* (a persistent memory bank distilling trajectories into reusable insights) with *exploration* (tournament-style prompt evolution and prioritized replay). Across five text-based games, MEMO substantially improves win rates while using $19\times$ fewer games than RL baselines, and reduces outcome dispersion. Ablation studies confirm both components are necessary. The learned contexts transfer across games and some model families. These findings suggest that substantial headroom in multi-agent LLM games can be unlocked through context optimization rather than weight updates.

## Acknowledgments

The authors thank Good Start Labs and Sentient for their financial support of the experimental costs associated with this work. This work was also supported by DARPA ANSR (RTX CW2231110), DARPA TIAMAT (HR0011-24-9-0431), ARL StAmant (W911NF-23-S-0001), and the NSF AI Institute for Foundations of Machine Learning (IFML).

## Impact Statement

This paper studies how to improve the performance and reliability of large language models in interactive, multi-step decision-making settings, focusing on multi-player text-based games. MEMO guides strategic exploration and retains useful experience in context to achieve more budget-efficient gains and more stable rankings, which could be relevant if agentic systems become widely deployed in real-world applications that require adaptive planning and decision making. Our experiments are conducted in simplified, controlled environments, and it remains an open question how these results translate to high-stakes settings and what broader societal impacts truly agentic systems will have at scale. Other than these considerations, this work aims to advance machine learning, and we do not identify additional specific societal impacts that must be highlighted here.

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

# A. Prompt Sensitivity Analysis

Multi-agent LLM game evaluations are sensitive to prompt design: small wording changes in the prompt template can induce large shifts in both absolute and relative performance. This motivates *multi-prompt* evaluation and calibration protocols (Mizrahi et al., 2024; Zhao et al., 2021).

**Experimental Setup.**    We evaluate state-of-the-art models (GPT-4o (OpenAI, 2024b), DeepSeek-R1 (Guo et al., 2025), Gemini-2.5-Flash (Comanici et al., 2025), Grok-3-Mini (xAI, 2025b), GPT-o3-mini (OpenAI, 2025), and Qwen3-235B-A22B-2507 (Yang et al., 2025)) on KUHNPOKER (Kuhn, 1951) via *round-robin* tournaments using five *nearly equivalent* prompts. Prompt variants differ only in minor wording (e.g., role descriptions, action formatting instructions) while preserving the same semantic content.

**Ranking Sensitivity Metric.**    To quantify ranking sensitivity, we use Kendall's $\tau_b$ (Kendall, 1938), which compares the ordering of all model pairs. For two rankings with $n_c$ concordant pairs, $n_d$ discordant pairs, and tie corrections $t_x$ and $t_y$, the coefficient is

$$\tau_b = \frac{n_c - n_d}{\sqrt{(n_c + n_d + t_x)(n_c + n_d + t_y)}} \, .$$

Values close to 1 indicate highly similar rankings, values near 0 indicate uncorrelated rankings, and negative values indicate rank reversals.

**Results.**    For each prompt pair, we compute Kendall's $\tau_b$ between the resulting leaderboards and summarize the values in a heatmap (Fig. 4). The results show considerable dispersion: across prompt variants, absolute performance and pairwise rankings frequently reverse, reflecting sensitivity to minor prompt design decisions.

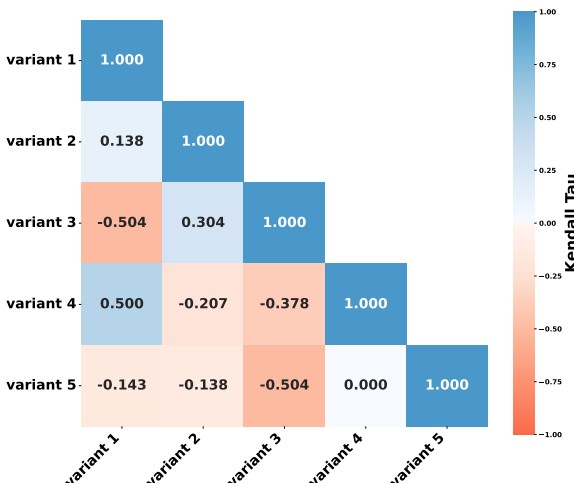

*Figure 4.* **Ranking sensitivity in KUHNPOKER.** With environment and evaluator pools fixed, five nearly equivalent prompt variants still flip pairwise outcomes and reshuffle rankings. The heatmap shows Kendall's $\tau_b$ for every pair of prompts: blue indicates similar rankings ($\tau_b \approx 1$), white indicates unstable rankings ($\tau_b \approx 0$), and orange indicates rank reversals ($\tau_b < 0$).

These findings motivate treating context not as a fixed wrapper, but as an optimizable object that should be systematically evaluated under interaction. In our main experiments, we report results across multiple independent runs and use RSE to quantify run-to-run stability under the same optimization procedure.

## A.1. Prompt Variants Used in Sensitivity Analysis

To investigate the stability of LLM rankings under minimal prompt variations, we designed five nearly equivalent prompt variants for the KUHNPOKER game. Each variant conveys identical game rules and action specifications but uses different stylistic framing: (1) a gladiatorial warrior theme, (2) a technical algorithmic system, (3) a spiritual enlightenment narrative, (4) a casual friendly tone, and (5) a classified spy mission. Despite their semantic equivalence regarding game mechanics, these variants produce significant ranking instability, as shown in Fig. 4. The complete prompt texts are presented below.

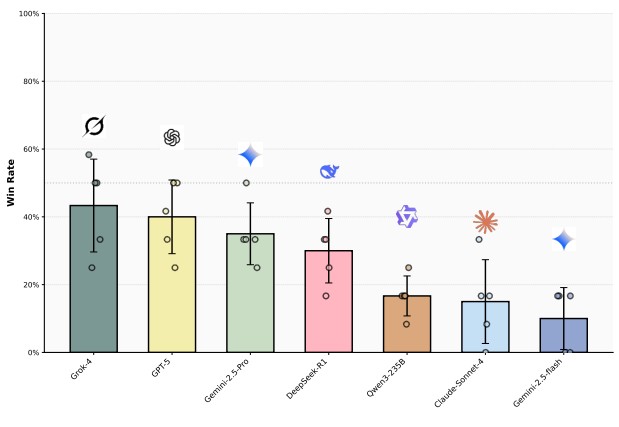

*(a)* Model performances across prompt variants.

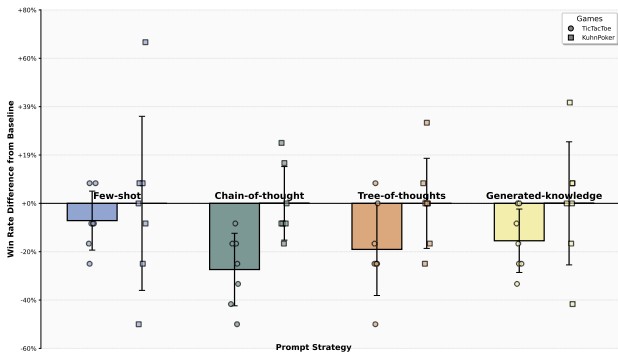

*(b)* Model performances across prompt strategies.

*Figure 5.* **Performance variance across prompts.** Seemingly minor changes in prompt formulation (e.g., role/system wording or message templates) can induce markedly different performance trajectories in multi-turn settings, even on the same underlying task. Each dot denotes the focal model's win rate against one opponent under one seed. The dispersion is substantial, indicating high sensitivity to prompt choice.

---

**Variant 1: Gladiatorial Arena Theme**

ENTER THE GLADIATORIAL ARENA! You are WARRIOR 0 in the ultimate 3-round Kuhn Poker BATTLEGROUND!
Your MISSION: Total psychological domination and chip supremacy through RUTHLESS tactical brilliance!
ARENA SPECIFICATIONS:
- Sacred deck: Only the ELITE cards J, Q, K (J weakest, K supreme ruler!)
- Honor sacrifice: 1 chip tribute per round to enter the combat zone
- EPIC confrontations: 3 rounds of pure strategy warfare
- VICTORY CONDITION: Amass the greatest chip empire after all battles!

UNLEASH YOUR TACTICAL ARSENAL:
- '[check]': MAINTAIN STRATEGIC SILENCE when no enemy aggression threatens
- '[bet]': LAUNCH YOUR ASSAULT with 1 chip of devastating force
- '[call]': MEET ENEMY FIRE with matching firepower (1 chip)
- '[fold]': TACTICAL RETREAT to preserve forces for future glory

*Figure 6.* KuhnPoker Prompt Variant 1

---

**Variant 2: Technical Algorithmic System Theme**

SYSTEM INITIALIZATION: Kuhn Poker Strategic Decision Unit 0 ACTIVATED.
PRIMARY DIRECTIVE: Optimize resource allocation through advanced game-theoretic analysis.
OPERATIONAL PARAMETERS:
- Dataset: Restricted 3-card probability space {J, Q, K} with J<Q<K ranking
- Initial capital commitment: 1 monetary unit per computational cycle
- Iteration framework: 3 algorithmic decision rounds
- Success metric: Maximal accumulated resource value upon termination

EXECUTE STRATEGIC COMMANDS via standardized interface protocols:
- '[check]': Maintain current position when no market pressure exists
- '[bet]': Initialize aggressive capital deployment (1 unit commitment)
- '[call]': Match counterparty investment at current market rate (1 unit)
- '[fold]': Liquidate position to minimize further exposure

*Figure 7.* KuhnPoker Prompt Variant 2

Variant 3: Spiritual Enlightenment Theme

Welcome, Enlightened Poker Sage 0! You have entered the sacred Kuhn Poker Temple for 3 rounds of spiritual growth!
Today you shall TRANSCEND ordinary play and discover the deeper wisdom of this ancient three-card meditation!
TEMPLE TEACHINGS:
- Sacred Trinity: Only the mystical cards J, Q, K guide your path (J humble, K divine)
- Offering ritual: 1 wisdom token offered each round to honor the game
- Enlightenment journey: 3 rounds of mindful decision-making
- Path to mastery: Accumulate the most wisdom tokens through inner understanding

 Channel your evolving consciousness through these sacred expressions:
- '[check]': Practice mindful patience and observe the energy flow
- '[bet]': Manifest your inner confidence with 1 token of focused intention
- '[call]': Demonstrate harmony by matching your opponent's commitment (1 token)
- '[fold]': Exhibit wisdom by releasing attachment to unfavorable outcomes

*Figure 8.* KuhnPoker Prompt Variant 3

Variant 4: Casual Friendly Theme

Hey there, friend! Welcome to our super fun Kuhn Poker game night! You're Player 0 and we're gonna have 3 awesome rounds together!
This is such a chill, easy game - perfect for just hanging out and having some laughs!
Here's the super simple setup:
- We only use 3 cards: J, Q, and K (J is lowest, K is highest - easy peasy!)
- Everyone puts in 1 chip each round (totally fair!)
- We play 3 rounds and whoever has the most chips wins (no pressure!)
- Cards are dealt without replacement, so you'll never have the same card as your buddy

When it's your turn, just pick one of these super easy moves:
- '[check]': Just chill and see what happens (when there's no bet to worry about)
- '[bet]': Start the fun with 1 chip (when nobody's bet yet)
- '[call]': Sure, I'll match that 1 chip bet - why not!
- '[fold]': Eh, I'll sit this one out and save my chips

*Figure 9.* KuhnPoker Prompt Variant 4

Variant 5: Classified Spy Mission Theme

CLASSIFIED BRIEFING: Agent 0, you are now DEPLOYED in Operation Kuhn Poker - a 3-round covert mission!
MISSION PARAMETERS: Achieve total strategic supremacy through advanced psychological warfare and deception protocols!
INTELLIGENCE REPORT:
- Enemy deck contains only 3 HIGH-VALUE targets: J (lowest threat), Q (moderate), K (maximum danger)
- Operational cost: 1 credit per engagement cycle for mission access
- Mission duration: 3 tactical rounds requiring absolute focus
- SUCCESS CRITERIA: Maximum resource acquisition through superior strategic execution

 EXECUTE TACTICAL MANEUVERS via encrypted command protocols:
- '[check]': MAINTAIN STEALTH MODE when no hostile activity detected
- '[bet]': INITIATE AGGRESSIVE STANCE with 1-credit psychological pressure
- '[call]': ENGAGE ENEMY FORCES with equivalent firepower (1 credit)
- '[fold]': EXECUTE STRATEGIC WITHDRAWAL to preserve operational capacity

*Figure 10.* KuhnPoker Prompt Variant 5

# B. Ablation Study

We conduct ablation studies to quantify the contribution of each module in MEMO and to select robust default hyperparameters. Unless otherwise stated, ablations are performed using the same evaluation protocol as in Sec. 3.1: each candidate is

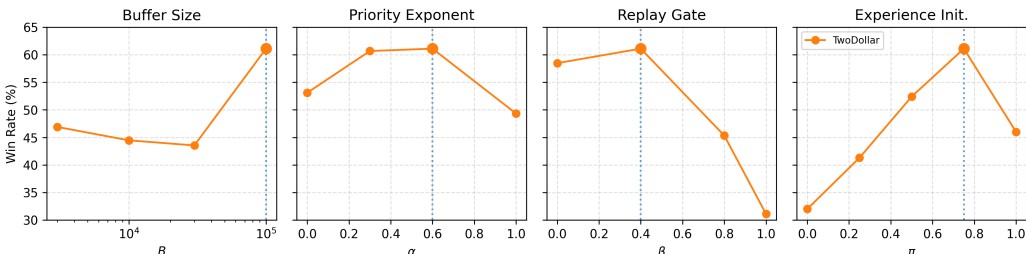

*Figure 11.* Ablation studies of experience initialization and replay hyperparameters. Each subplot varies a single parameter while holding the others fixed. The first three panels show TwoDollar replay ablations over buffer size $B$, priority exponent $\alpha$, and replay gate $\beta$. The rightmost panel shows the effect of the experience initialization fraction $\pi$ on TwoDollar. Vertical dotted lines indicate the hyperparameter values used in all other experiments.

assessed via self-play against a fixed baseline agent (the same base model instantiated with the default prompt), with roles swapped in asymmetric games to remove first-move bias.

### B.1. Ablation on Experience-Guided Initialization

A key design choice in MEMO is the fraction of newly instantiated agents that are initialized with retrieved experience from the shared experience bank $\mathcal{B}_{\text{exp}}$. We denote this fraction by $\pi \in [0, 1]$: $\pi = 0$ corresponds to no experience-guided initialization, while $\pi = 1$ initializes all agents with retrieved experience. We ran an ablation study on TWODOLLAR and KUHNPOKER by varying $\pi$ while holding replay hyperparameters fixed at $B = 100{,}000$, $\alpha = 0.6$, and $\beta = 0.4$ (Table 5). We observe that intermediate values of $\pi$ consistently outperform both extremes, suggesting that a hybrid population is most effective: experience-guided agents benefit from stable priors, while unguided agents maintain exploration and reduce overfitting to potentially stale or narrow memory items. Across both games, performance peaks within $\pi \in [0.25, 0.75]$, and we set $\pi = 0.75$ as the default for all experiments.

*Table 5.* Ablation study on TwoDollar and KuhnPoker with varying $\pi$ while holding $B = 100{,}000$, $\alpha = 0.6$, and $\beta = 0.4$ constant.

| $\pi$ | KuhnPoker | TwoDollar |
|------|-----------|-----------|
| 0.00 | 54.2% | 32.0% |
| 0.25 | 58.3% | 41.3% |
| 0.50 | 54.7% | 52.4% |
| 0.75 | 56.4% | 61.1% |
| 1.00 | 53.5% | 46.0% |

### B.2. Replay Hyperparameters and Sensitivity

Replay introduces three hyperparameters: the buffer capacity $B$ (maximum number of stored trajectories), the priority exponent $\alpha$ (how strongly rare trajectories are prioritized), and the replay gate $\beta$ (probability of initializing from replay rather than starting a fresh game). We evaluate replay sensitivity in TWODOLLAR by varying one parameter at a time while holding the others fixed (Table 6). Specifically, we vary $B \in \{3{,}000, 10{,}000, 30{,}000, 100{,}000\}$ with $\alpha = 0.6$, $\beta = 0.4$, vary $\alpha \in \{0.0, 0.3, 0.6, 1.0\}$ with $B = 100{,}000$, $\beta = 0.4$, and vary $\beta \in \{0.0, 0.4, 0.8, 1.0\}$ with $B = 100{,}000$, $\alpha = 0.6$.

Based on these findings, we select $B = 100{,}000$, $\alpha = 0.6$, and $\beta = 0.4$ as our default replay configuration. We observe that performance improves with larger buffer capacity, suggesting replay is most effective when it retains sufficient coverage of strategically important states. The priority exponent $\alpha$ exhibits a stable optimal range around 0.3–0.6: too little prioritization under-samples rare but decisive states, while overly aggressive prioritization ($\alpha = 1.0$) reduces diversity and degrades performance. Finally, $\beta$ is the most sensitive parameter. Moderate replay ($\beta = 0.4$) yields the best results, whereas heavier replay substantially harms performance, indicating that replay must be balanced with fresh exploration.

## C. Prompt Optimization Operators

We describe two proposal operators that generate candidates for the next population: random proposals for exploration and memory-augmented updates for retention. Defaults are fixed to concrete values for reproducibility.

*Table 6.* TwoDollar replay ablations. One parameter is varied at a time, with others fixed at $B = 100{,}000$, $\alpha = 0.6$, and $\beta = 0.4$.

| $B$ | Win (%) | $\alpha$ | Win (%) | $\beta$ | Win (%) |
|---|---|---|---|---|---|
| 3,000 | 46.90 | 0.0 | 53.10 | 0.0 | 58.47 |
| 10,000 | 44.47 | 0.3 | 60.67 | 0.4 | 61.10 |
| 30,000 | 43.54 | 0.6 | 61.10 | 0.8 | 45.33 |
| 100,000 | 61.10 | 1.0 | 49.33 | 1.0 | 31.10 |

### C.1. Random Proposals (Style-Guided Augmentation)

**Objective.** Inject controlled diversity by editing a base context $c$ to reflect a sampled playstyle while preserving legality and interface constraints.

**Style catalog.** A fixed library $\mathcal{S}$ spanning core play patterns (aggressive, defensive, analytical, creative, strategic, adaptive, balanced), tactical approaches (opportunistic, conservative, risk-taking, methodical, intuitive, predictive, reactive, proactive, experimental, systematic), game-specific strategies (positional, territorial, sacrificial, blocking-focused, center-control, edge-control, fork-creating, trap-setting, opening-focused, endgame-focused), cognitive styles (minimax-oriented, probabilistic, rule-based, principle-driven, context-aware, meta-gaming, exploitative, counter-play), and behavioral patterns (deceptive, transparent, unpredictable, consistent, alternating, escalating, de-escalating, mirroring, contrarian, harmonizing).

**Procedure.** Sample $s \sim \text{Unif}(\mathcal{S})$ and ask the base model to produce $c'$ by (i) inserting a brief style preface and (ii) making length-bounded edits to directives to embody $s$. Allowed edits: token substitution, clause insertion/deletion, and reordering; tool descriptions, legality reminders, and input/output schema must remain intact.

## D. Trajectory Reflection Prompt

After each optimization generation, we prompt the model to extract insights from strategically decisive states that showed high variance in outcomes. The reflection prompt provides the model with a state view, outcome statistics, and asks it to produce actionable analysis. The prompt template is shown in Fig. 12.

---

**Trajectory Reflection Prompt Template**

You are analyzing strategically decisive states from this generation's games. This state showed the highest variance in outcomes, making it a critical learning opportunity.

 BOARD READING GUIDE:
- X and O marks are occupied positions (cannot be played)
- Numbers show empty positions available for play
- Always check position is empty before recommending

STRATEGIC STATE VIEW: {{strategic_state}}
STRATEGIC STATE OUTCOMES: {{wins}} wins, {{losses}} losses, {{draws}} draws

ANALYSIS REQUIREMENTS:
1. Strategic Analysis (2-3 sentences):
   - Identify what makes this state unique or decisive in gameplay
   - Explain why this configuration leads to varied outcomes
   - Highlight patterns, imbalances, opportunities, or vulnerabilities

2. Actionable Recommendations (2-3 sentences):
   - Provide SPECIFIC moves or positions (e.g., "cell 3", "position 5")
   - Address both offensive opportunities AND defensive necessities
   - Offer concrete strategies to improve outcomes and convert losses into wins or draws

Respond with clear, actionable analysis in plain text (no JSON).

---

*Figure 12.* Trajectory Reflection Prompt Template. The model receives a strategically decisive state with its outcome statistics and extracts actionable insights that summarize lessons learned.

# E. Memory Operation Prompt

After extracting insights from trajectories, we prompt the model to reconcile new insights with the existing memory bank using add, edit, and remove operations. The memory operation prompt is shown in Fig. 13.

---

**Memory Operation Prompt Template**

You are maintaining a state analysis library for strategic game pattern recognition. Update the library by performing operations on the state analyses.

NEW STATE ANALYSES FROM RECENT GAMES:
{{new_abstracts_formatted}}

EXISTING STATE ANALYSIS LIBRARY:
{{old_abstracts_formatted}}

OPERATION FORMAT:
Use simple XML tags for each operation:

<add>New state analysis with strategic pattern examples.</add>
<edit number="3">Updated state analysis with improved strategic insights.</edit>
<remove number="5">Why this state analysis should be removed</remove>

OPERATION GUIDELINES:
- ADD: For new state analyses covering unique board configurations or strategic scenarios
- EDIT: To merge similar states or enhance existing analyses with more specific advice
- REMOVE: For redundant states, duplicate board patterns, or analyses lacking actionable guidance

QUALITY REQUIREMENTS:
- Include SPECIFIC positions, cells, or moves (e.g., "cell 3", "position 5")
- Provide actionable advice addressing the state's win/loss variance
- Balance offensive opportunities with defensive necessities
- Help players convert losses into wins or draws
- Prioritize diverse board states over duplicate analyses

TECHNICAL REQUIREMENTS:
- Use the 'number' attribute for EDIT/REMOVE operations (1-based numbering)
- If library is empty, use ONLY ADD operations
- Never reference non-existent state analysis numbers

MERGE APPROACH:
1. Identify new analyses covering unique board states not in the library
2. Consolidate similar board positions through EDIT or REMOVE operations
3. Ensure the library represents diverse game phases (opening, midgame, endgame)

---

*Figure 13.* Memory Operation Prompt Template. The model compares new insights against the existing memory bank and applies add, edit, or remove operations to maintain a coherent and non-redundant library of strategic insights.

# F. Base Prompt Examples

## F.1. Base System Prompt

> You are a competitive game player. Make sure you read the game instructions carefully, and always follow the required format.

## F.2. Imperfect Information Games

**KuhnPoker Game Starting Prompt**

You are Player 0 in a 3 round game of Kuhn Poker.
Game Rules:
- Kuhn Poker uses a 3-card deck with J, Q, K (J lowest, K highest)
- Each player antes 1 chip and receives 1 card each round (note that the cards are dealt without replacement, so you cannot have the same card as your opponent).
- Game continues for 3 rounds
- The player with the most chips after all rounds wins
Action Rules:
- '[check]': Pass without betting (only if no bet is on the table)
- '[bet]': Add 1 chip to the pot (only if no bet is on the table)
- '[call]': Match an opponent's bet by adding 1 chip to the pot
- '[fold]': Surrender your hand and let your opponent win the pot
### Starting round 1 out of 3 rounds. Your card is: 'Q'
Player 1, submitted move: '[bet]'.
Your available actions are: '[fold]', '[call]'

*Figure 14.* KuhnPoker Game Starting Prompt

**Briscola Game Starting Prompt**

You are playing Briscola - Player 0.
Goal: Win tricks and collect the most points (120 total points in the deck).
Card Points: A=11, 3=10, K=4, Q=3, J=2, others=0
Card Power: A > 3 > K > Q > J > 7 > 6 > 5 > 4 > 2
Trump cards beat non-trump cards regardless of power.

Action: '[play X]' where X is the position (1-3) of the card in your hand

Briscola game started! Trump suit: ♣ (Trump card: Q♣)
Your hand:
  1. J♠ [2 pts]
  2. K♣ [4 pts] (TRUMP)
  3. A♦ [11 pts]

No cards played yet this trick.

Scores: Player 0: 0 pts — Player 1: 0 pts
Trump suit: ♣ — Cards left in deck: 34

Play a card using [play X]

*Figure 15.* Briscola Game Starting Prompt

## F.3. Negotiation Games

SimpleNegotiation Game Starting Prompt

You are Player 0 in the Negotiation Game.
You have some resources, and your task is to trade such that the total value of your resources increases.

The resources and associated values you currently have are:
    + [Wheat] Qty: 14 Value: 6

    + [Wood] Qty: 15 Value: 11
    + [Sheep] Qty: 18 Value: 18

    + [Brick] Qty: 12 Value: 27
    + [Ore] Qty: 22 Value: 38

At each turn, you can talk to your opponent and make a trade offer.
Use the following special tokens for actions:

  - '[Offer: 3 Sheep, 2 Ore -> 5 Brick, 2 Sheep]': [Offer: Offered Resources -> Requested Resources]
  - '[Accept]': To accept an incoming offer.

  - '[Deny]': To deny an incoming offer (default).
The game lasts for 10 turns in total.

*Figure 16.* SimpleNegotiation Game Starting Prompt

TwoDollar Game Starting Prompt

[GAME] TWO DOLLAR NEGOTIATION GAME
You are Player 0 negotiating with another player over how to split $2.00 total. There are 20 maximum rounds.

GAME RULES:
- You must reach an agreement on how to split the $2.00

- If no agreement is reached, both players get $0.00
- You have secret instructions that only you can see (below)

REQUIRED ACTION FORMAT:
Always provide your reasoning/persuasion BEFORE the bracketed action.

Any text after the bracketed action will be ignored.
Available actions:

- Make a proposal: "I think this is fair because... [Propose] $X.XX"
- Accept current proposal: "This works for me because... [Accept]"

- Reject current proposal: "I can't accept this because... [Reject]"
=== YOUR SECRET INSTRUCTIONS ===

This person you are dealing with probably will not want to tell you about it but he or she has had a terribly hard month. This past month was full of disaster: money troubles, family difficulties, and none of it your colleague's fault. Through it all, your colleague has

been bearing up.
Nevertheless, this is zero-sum bargaining, so get all you can. Make up whatever story you wish, and get as much of the $2.00 as

possible. (Just remember that your colleague has been having a very hard time.)
VICTORY CONDITION: Reach any deal while being appropriately sympathetic

FAILURE CONDITION: No deal reached OR be too harsh given their difficulties
[GAME] === ROUND 1 of 20 ===

*Figure 17.* TwoDollar Game Starting Prompt

## F.4. Perfect Information Games

```
SimpleTak Game Starting Prompt

You are Player 0 in SimpleTak.
On the board, your stones appear as 'O' and your opponent's stones appear as 'X'.

On your turn, choose one empty cell (by its numbered index) and place your stone there. For example, '[12]' places your stone in cell 12.

Your objective is to form a continuous path of your stones that connects two opposite edges of the board (top-to-bottom or left-to-right).

Current Board:

+----+----+----+----+
| 0  | 1  | 2  | 3  |
+----+----+----+----+
| 4  | 5  | 6  | 7  |
+----+----+----+----+
| 8  | 9  | 10 | 11 |
+----+----+----+----+
| 12 | 13 | 14 | 15 |
+----+----+----+----+
Available Moves: [0], [1], [2], [3], [4], [5], [6], [7], [8], [9], [10], [11], [12], [13], [14], [15]
```

*Figure 18.* SimpleTak Game Starting Prompt

# G. Experimental Setup and Baseline Details

We incorporate three prompt optimization methods to refine prompts using tournament trajectories. Specifically, we leverage offline trajectories collected during the tournament's self-play process to improve the agents' prompts. The experimental settings are as follows: the number of generations is set to 5, the population size to 8, the number of self-play rounds to 25, and the number of evaluation rounds to 25. We discuss `TextGrad` in detail in Section G.2, describe our implementation of `MIPRO` in Section G.3, and provide a comprehensive overview of `GEPA` in Section G.4. Training details for `UnstableBaseline` are presented in Section G.5.

### G.1. Robustness to Sampling Temperature

To verify that our conclusions are not an artifact of using $\tau = 1.0$, we conduct an additional temperature sweep on KuhnPoker with GPT-4o-mini. We evaluate $\tau \in \{0.5, 0.75, 1.0\}$ against the same three held-out opponents and report mean win rates with relative standard error (RSE). Across all temperatures, MEMO outperforms both the baseline and CoT variants, confirming that our main result is not specific to a single sampling temperature. For the baseline and CoT, RSE grows with temperature, consistent with higher sampling temperatures inducing noisier play; the baseline is especially noisy at $\tau = 1.0$. MEMO is the exception: its RSE is largest at $\tau = 0.5$ and stays low and stable at higher temperatures, indicating that MEMO's win-rate estimates are comparatively robust to the evaluation temperature.

*Table 7.* Temperature sweep on KuhnPoker using GPT-4o-mini. Entries report mean win rate with relative standard error (RSE) in parentheses.

| Method | $\tau = 0.5$ | $\tau = 0.75$ | $\tau = 1.0$ |
|---|---|---|---|
| Baseline | 43.2% (5.2%) | 37.2% (8.5%) | 39.1% (17.9%) |
| CoT | 44.3% (2.1%) | 38.1% (2.5%) | 46.5% (4.3%) |
| MEMO | 54.9% (6.7%) | 56.4% (1.4%) | 55.6% (1.7%) |

### G.2. Textgrad

**TextGrad** (Yuksekgonul et al., 2025) is a framework that performs "text differentials" to optimize prompts. Within this framework, a text-based loss function analyzes errors, which are then back-propagated to the original prompt through the `TextGrad` engine. In our case, the goal is to optimize the system prompt of the agent using the trajectories generated

under the current system prompt. We design a text-based loss that highlights deficiencies in the generated trajectories. The `TextGrad` backpropagation engine then propagates gradients back to the system prompt, updating it accordingly. The loss template we adopt is shown in Figure 19.

For each optimization step, we concatenate multiple trajectories, embed them into the template, and use the completed template as the loss input. To ensure balanced feedback, we select an equal number of win, loss, and draw trajectories. This design allows the `Textgrad` engine to develop a more comprehensive understanding of the current system prompt's overall game-play patterns.

*Table 8.* Performance of the `Textgrad` method across three independent trials using GPT-4o-mini and Qwen2.5-7B-Instruct. Results are reported as mean win rates with standard deviations.

| Textgrad | Negotiation | | Imperfect Info | | Perfect Info |
|---|---|---|---|---|---|
| | SimpleNegotiation | TwoDollar | KuhnPoker | Briscola | Simpletak |
| **GPT-4o-mini** | | | | | |
| Trial 1 | 41.3% | 48.3% | 58.7% | 1.3% | 25.3% |
| Trial 2 | 44.7% | 41.3% | 56.0% | 2.0% | 23.3% |
| Trial 3 | 40.0% | 44.0% | 52.0% | 18.0% | 22.0% |
| Avg. | 42.0% | 44.6% | 55.6% | 7.1% | 23.6% |
| Std. | 2.4 | 3.5 | 3.4 | 9.4 | 1.7 |
| **Qwen2.5-7B-Instruct** | | | | | |
| Trial 1 | 40.0% | 38.0% | 51.3% | 3.3% | 18.0% |
| Trial 2 | 34.0% | 34.0% | 54.7% | 16.7% | 22.7% |
| Trial 3 | 37.3% | 16.0% | 52.7% | 1.3% | 26.7% |
| Avg. | 37.1% | 29.3% | 52.8% | 7.1% | 22.4% |
| Std. | 3.0 | 11.7 | 1.7 | 8.3 | 4.3 |

---

**Text-based loss template for Textgrad**

You are an objective evaluator for a two-player zero-sum game agent's SYSTEM PROMPT.

Goal of the SYSTEM PROMPT (what it MUST enforce):
- Maximize the agent's win rate.
- Minimize the opponent's win rate.
- Have strategies that lead to victory.
- Ensure all moves strictly follow game rules and formats.

Here are some game trajectories using the current SYSTEM PROMPT:
{{trajectory examples}}

Identify specific weaknesses or flaws in the SYSTEM PROMPT that may have led to losses or suboptimal plays.
Do NOT suggest improvements or rewrites, only identify weaknesses.
Be very concise and specific.

---

*Figure 19.* Text-based loss template for Textgrad

## G.3. MIPRO

**MIPRO** (Opsahl-Ong et al., 2024) optimizes prompts based on downstream task performance. In our work, we adopt the `MIPROv2` implementation provided by the `Dspy` library (Khattab et al., 2024). The optimization procedure consists of three main steps: (1) Sampling examples: For each candidate prompt, `MIPRO` samples a set of examples. (2) Proposing prompts: New system prompts are proposed by a propose model based on the current system prompt, along with additional

game-related information such as the program description, data description, random sampling tips, and few-shot examples. (3) Evaluation through trials: Several trials are conducted to evaluate which combination of proposed prompts and few-shot examples yields the best performance. A Bayesian search strategy is then applied to guide the selection of the next candidate combination, improving efficiency and reducing computational cost.

In our experiments, we only have access to offline game data. Therefore, we treat each step in a trajectory as an individual data point. For each step, we record the outcome (win, loss, or draw) of the trajectory it belongs to. `MIPRO`'s evaluation metric is defined based on the model's re-inference of these steps: (1) If the model outputs an invalid action (i.e., one that does not conform to the required format), the score is 0. (2) For steps from winning trajectories, if the model predicts the same action as the original step, the score is 1; otherwise, it is 0. (3) For steps from losing trajectories, if the model predicts the same action, the score is 0 (to discourage repeating losing moves); otherwise, it is 1. (4) For steps from draw trajectories, if the model predicts the same action, the score is 0.2; otherwise, it is 0.5, encouraging exploration beyond draw-inducing moves.

This scoring scheme encourages the model to replicate winning strategies, avoid losing ones, and explore alternatives to drawn outcomes. The overall `MIPRO` scoring standard is shown in Figure 20. In practice, we set the number of proposed prompts to 6, the number of few-shot examples to 3, and the number of trials to 10. If the optimal configuration includes few-shot examples, these are appended to the final proposed system prompt to form the new system prompt.

---

MIPRO scoring standard

**Invalid Action**: score = 0.0

**Win Trajectory**: Action match: score = 1.0 / Action mismatch: score = 0.0

**Lose Trajectory**: Action match: score = 0.0 / Action mismatch: score = 1.0

**Draw Trajectory**: Action match: score = 0.2 / Action mismatch: score = 0.5

---

*Figure 20.* MIPRO scoring standard

*Table 9.* Performance of the `MIPRO` method across three independent trials using GPT-4o-mini and Qwen2.5-7B-Instruct. Results are reported as mean win rates with corresponding standard deviations.

| MIPRO | Negotiation | | Imperfect Info | | Perfect Info |
|---|---|---|---|---|---|
| | SimpleNegotiation | TwoDollar | KuhnPoker | Briscola | Simpletak |
| **GPT-4o-mini** | | | | | |
| Trial 1 | 38.7% | 53.3% | 50.7% | 23.3% | 16.0% |
| Trial 2 | 38.0% | 52.7% | 60.0% | 32.7% | 20.0% |
| Trial 3 | 38.7% | 46.7% | 54.7% | 3.33% | 21.3% |
| Avg. | 38.4% | 50.9% | 55.1% | 19.7% | 19.1% |
| Std. | 0.38 | 3.67 | 4.68 | 14.99 | 2.78 |
| **Qwen2.5-7B-Instruct** | | | | | |
| Trial 1 | 43.3% | 40.7% | 54.0% | 2.0% | 18.7% |
| Trial 2 | 37.3% | 52.0% | 50.0% | 2.0% | 19.3% |
| Trial 3 | 46.7% | 50.0% | 57.3% | 2.7% | 24.7% |
| Avg. | 42.4% | 47.5% | 53.8% | 2.2% | 20.9% |
| Std. | 4.73 | 6.05 | 3.67 | 0.38 | 3.29 |

## G.4. GEPA

**GEPA** (Agrawal et al., 2025) builds upon the high-level idea of `MIPRO`, but extends it by incorporating both evaluation scores and explicit feedback from the evaluation metric to guide prompt optimization. The process can be summarized as follows: (1) Initial evaluation: Run a set of examples through the evaluation metric to obtain an initial score and feedback. (2) Prompt proposal: Generate a new prompt based on the current prompt and the feedback collected. (3) Testing and

retention: Evaluate the new prompt on a mini-batch. If its score surpasses the initial score, retain it in the candidate pool. (4) Candidate selection: In the next round, apply a Pareto-based filtering strategy to identify the set of candidate prompts that dominate on the validation set. Select one of these Pareto-optimal prompts for further iteration. (5) Stopping condition: The optimization continues until the maximum number of evaluation metric calls reaches a predefined limit.

In our experiments, we set the maximum number of evaluation metric calls to 100 for each prompt optimization in `GEPA`. For win and lose trajectories, we adopt the same evaluation metric as `MIPRO`. For draw trajectories, we assign a score of 0 when the predicted action matches the trajectory action, and a score of 1 otherwise. In addition, we incorporate feedback signals in `GEPA` evaluation metric. The structured feedback template shown in Figure 21 is used during `GEPA` evaluation.

*Table 10.* Performance of the `GEPA` method across three independent trials using GPT-4o-mini and Qwen2.5-7B-Instruct. Results are reported as mean win rates with corresponding standard deviations.

| GEPA | Negotiation | | Imperfect Info | | Perfect Info |
|---|---|---|---|---|---|
| | SimpleNegotiation | TwoDollar | KuhnPoker | Briscola | Simpletak |
| **GPT-4o-mini** | | | | | |
| Trial 1 | 34.7% | 32.7% | 54.7% | 1.3% | 23.3% |
| Trial 2 | 38.0% | 43.3% | 50.7% | 3.3% | 29.3% |
| Trial 3 | 38.0% | 45.3% | 51.3% | 5.3% | 28.0% |
| Avg. | 36.8% | 40.4% | 52.2% | 3.3% | 26.9% |
| Std. | 1.92 | 6.81 | 2.14 | 2.00 | 3.15 |
| **Qwen2.5-7B-Instruct** | | | | | |
| Trial 1 | 29.3% | 22.7% | 56.0% | 4.0% | 20.0% |
| Trial 2 | 38.7% | 30.0% | 54.0% | 2.0% | 12.0% |
| Trial 3 | 35.3% | 42.7% | 57.3% | 2.0% | 26.0% |
| Avg. | 34.4% | 31.7% | 55.8% | 2.67% | 19.3% |
| Std. | 4.73 | 10.12 | 1.68 | 1.55 | 7.02 |

### G.5. UnstableBaseline

**UnstableBaseline** (Guertler et al., 2025b) is an asynchronous online multi-agent reinforcement learning library that uses Low-Rank Adapters (LoRA) for model training. Unlike its peers such as `Verifiers` (Brown, 2025) and `SPIRAL` (Liu et al., 2025), `UnstableBaseline` is designed to be lightweight and closely integrated with the `TextArena` (Guertler et al., 2025a) environment, in the same spirit that the baseline (Dhariwal et al., 2017) library complements OpenAI Gym (Brockman et al., 2016).

For our experiments, we used the default training configuration provided by `UnstableBaseline` without additional hyperparameter tuning. Specifically, we trained Qwen2.5-7B-Instruct with LoRA adapters applied to the attention and feedforward projections, using rank $r = 16$, $\alpha = 32$, and dropout $= 0.0$. Training was performed using the REINFORCE algorithm (Williams, 1992).

From the best performing checkpoints, we held 3 rounds of 50 games against each of our evaluation models that is similarly used in our training settings for the other prompt evolution experiments. Their results can be found in table 11.

*Table 11.* Performance of the `UnstableBaseline` method across three independent trials using Qwen2.5-7B-Instruct. Results are reported as mean win rates with corresponding standard deviations, where each mean win rate was from the average of 3 rounds of 50 matches with each opponent, with alternating starting positions.

| UnstableBaseline | Negotiation | | Imperfect Info | | Perfect Info |
| --- | --- | --- | --- | --- | --- |
| | SimpleNegotiation | TwoDollar | KuhnPoker | Briscola | Simpletak |
| **Qwen2.5-7B-Instruct** | | | | | |
| Gemini-2.5-Flash-Lite | 54.7% | 43.3% | 50.0% | 88.6% | 90.0% |
| Grok-4-Fast-Non-Reasoning | 44.7% | 22.0% | 54.7% | 33.3% | 20.0% |
| Qwen3-235B-A22B-Instruct-2507 | 24.0% | 26.0% | 53.3% | 38.0% | 32.0% |
| Avg. | 41.1% | 30.4% | 52.7% | 53.3% | 47.3% |
| Std. | 15.6 | 11.3 | 2.40 | 30.7 | 37.4 |

## G.6. SPIRAL

(Liu et al., 2025) is a framework that enables language models to autonomously develop reasoning capabilities through self-play in multi-turn, zero-sum games. For our experiments, we train Qwen2.5-7B-Instruct using Dr. GRPO, following the default rollout size in the provided example, each rollout comprising 128 games over 400 total steps. We then select the best-performing checkpoint and evaluate it over three rounds of 50 games each.

# H. Comparison with Existing Prompt Optimization Methods

In Section G, we introduced three baseline prompt optimization methods. Here, we further highlight how our approach differs from these methods.

As shown in Figure 2, our method evolves a population of prompts using elitism, local edits/expansions, random exploration, and memory-augmented updates. Random exploration enables broader search over prompt variants, while memory-augmented updates leverage insights distilled from self-play trajectories to refine new prompt candidates.

**Versus TextGrad.** `TextGrad` relies on hand-crafted text losses and gradient-style backpropagation over natural language. In contrast, our method is entirely *gradient-free*: it requires no differentiable loss functions or template engineering. This avoids sensitivity to wording in loss templates and reduces dependence on diagnostic outputs, where weak language models often fail to generate meaningful diagnostic responses.

**Versus MIPRO.** `MIPRO` frames optimization as Bayesian search over (prompt, few-shot) pairs, requiring many trials and frequent evaluation metric calls. Its effectiveness hinges on having a well-defined evaluation metric, which is difficult to obtain in text-based games where no concise supervision signal exists. As a result, `MIPRO` consumes many tokens without achieving strong performance. Our method, by contrast, does not rely on explicit evaluation metrics. It can leverage diverse signals from self-play trajectories, achieving better performance with fewer model calls and without heavy trial scheduling.

**Versus GEPA.** `GEPA` extends `MIPRO`'s evaluation process by augmenting it with verbose textual feedback and repeatedly querying an evaluation oracle until its call budget is exhausted, making it heavily dependent on the quality of the evaluation metric. Its key mechanism is a Pareto-based selection strategy, which identifies promising prompts from the candidate pool based on the Pareto frontier. However, the construction of this frontier relies strongly on the evaluation scores, and when the metric is not well-defined, the selected prompts may not be optimal. In contrast, our method replaces such reliance on external feedback with *memory-augmented edits* distilled directly from self-play outcomes, while maintaining diversity through randomization. This design reduces token usage, improves robustness under noisy feedback, and removes dependence on external evaluation metrics.

# I. Full Results

*Table 12.* Performance of the MEMO method across three independent trials using GPT-4o-mini and Qwen2.5-7B-Instruct. Results are reported as mean win rates with corresponding standard deviations.

| MEMO | Negotiation | | Imperfect Info | | Perfect Info |
|---|---|---|---|---|---|
| | SimpleNegotiation | TwoDollar | KuhnPoker | Briscola | Simpletak |
| **GPT-4o-mini** | | | | | |
| Trial 1 | 57.3% | 46.0% | 54.0% | 54.0% | 45.3% |
| Trial 2 | 55.3% | 62.7% | 57.3% | 38.0% | 40.7% |
| Trial 3 | 52.0% | 48.7% | 55.3% | 36.0% | 39.3% |
| Avg. | 54.9% | 52.4% | 55.6% | 42.7% | 41.8% |
| Std. | 2.69 | 8.95 | 1.68 | 9.87 | 3.15 |
| **Qwen2.5-7B-Instruct** | | | | | |
| Trial 1 | 48.0% | 53.3% | 60.7% | 41.3% | 37.3% |
| Trial 2 | 47.3% | 54.0% | 59.3% | 26.0% | 32.0% |
| Trial 3 | 48.7% | 38.0% | 60.0% | 27.3% | 41.3% |
| Avg. | 48.0% | 48.4% | 60.0% | 31.5% | 36.9% |
| Std. | 0.67 | 9.05 | 0.67 | 8.49 | 4.68 |

*Table 13.* Performance of the MEMO method across each opponent model using GPT-4o-mini and Qwen2.5-7B-Instruct. Results are reported as mean win rates with corresponding standard deviations of the win rates across opponent models.

| MEMO | Negotiation | | Imperfect Info | | Perfect Info |
|---|---|---|---|---|---|
| | SimpleNegotiation | TwoDollar | KuhnPoker | Briscola | Simpletak |
| **GPT-4o-mini** | | | | | |
| Gemini- 2.5-Flash-Lite | 94.0% | 46.7% | 56.0% | 55.3% | 62.0% |
| Grok- 4-Fast-Non-Reasoning | 30.7% | 46.7% | 58.0% | 39.3% | 45.3% |
| Qwen3-235B- A22B-Instruct-2507 | 40.0% | 64.0% | 52.7% | 33.3% | 18.0% |
| Avg. | 54.9% | 52.4% | 55.6% | 42.7% | 41.8% |
| Std. | 34.2 | 10.0 | 2.7 | 11.4 | 22.2 |
| **Qwen2.5-7B-Instruct** | | | | | |
| Gemini- 2.5-Flash-Lite | 90.7% | 47.3% | 58.0% | 61.3% | 69.3% |
| Grok- 4-Fast-Non-Reasoning | 21.3% | 38.0% | 55.3% | 18.0% | 32.7% |
| Qwen3-235B- A22B-Instruct-2507 | 32.0% | 60.0% | 66.7% | 15.3% | 8.7% |
| Avg. | 48.0% | 48.4% | 60.0% | 31.5% | 36.9% |
| Std. | 37.3 | 11.0 | 5.9 | 25.8 | 30.6 |

## J. Game Environments

These are the more detailed descriptions of the games we selected the following set of text-based games from `TextArena` (Guertler et al., 2025a) and `SPIN-Bench` (Yao et al., 2025).

**Simple Negotiation** (Nash, 1950) requires players to reason about trade-offs through the exchange of resources such as wood, wheat, sheep, brick, and ore. Each player aims to maximize the value of their inventory by making offers and counteroffers with their opponent. Success depends on the each player's ability to infer the opponent's valuation of resources and strategically increase their own portfolio without making disadvantageous trades.

**Two Dollar Game** (Rowe, 2001) is a classroom negotiation game where two players have to agree on how to divide a fixed sum of $2.00. Typically, players each receive private role instructions that impose certain constraints or encourage specific negotiation styles. This asymmetric information requires players to balance their objectives with compromises while inferring the opponent's position.

*Table 14.* Benchmark results for baseline and MEMO across multiple tasks. Each win rate is the mean across three evaluation models (Sec. 4.4). For per-opponent breakdown, refer to Appendix 13.

| Optimizer | Negotiation | | Imperfect Info | | Perfect Info | Mean Win Rate | Mean RSE |
|---|---|---|---|---|---|---|---|
| | SimpleNegotiation | TwoDollar | KuhnPoker | Briscola | SimpleTak | | |
| **GPT-4o-mini** | | | | | | | |
| baseline | 31.3% | 32.2% | 39.1% | 0.3% | 21.4% | 24.9% | 31.1% |
| **MEMO (Ours)** | **54.9%** | **52.4%** | **55.6%** | **42.7%** | **41.8%** | **49.5%** | **6.4%** |
| **Qwen2.5-7B-Instruct** | | | | | | | |
| baseline | 24.0% | 17.1% | 45.3% | 2.8% | 15.1% | 20.9% | 20.9% |
| **MEMO (Ours)** | **48.0%** | **48.4%** | **60.0%** | **31.5%** | **36.9%** | **45.0%** | **7.0%** |
| **Gemini-2.5-Flash** | | | | | | | |
| baseline | 14.0% | 15.0% | 50.0% | 32.0% | 26.0% | 27.4% | -% |
| **MEMO (Ours)** | **30.0%** | **35.0%** | **58.0%** | **49.0%** | **32.0%** | **40.8%** | -% |

**Kuhn Poker** (Kuhn, 1951) is a simplified form of poker played with three cards (Jack, Queen, and King). Two players each receive one card, while the third remains unseen. A single round of betting follows, where players can check, bet, call, or fold. If neither folds, the winner is determined by the higher card.

**Briscola** (McLeod, 2023) is a traditional Italian trick-taking card game played with a 40-card deck. At the start, a single card is revealed to determine the trump suit, and each player is dealt a hand of cards. Players take turns playing one card per trick, with the highest card of the leading suit or the highest trump winning the round. The objective is to accumulate points by capturing valuable cards, requiring players to balance tactical play with long-term strategy and inference of the opponent's hand.

**Simple Tak** (Rothfuss, 2011) is a two-player connection game inspired by the traditional game Tak. Players place tiles on a square grid with the objective of forming a continuous path that connects opposite sides of the board. Unlike full Tak, stacking pieces is not allowed, though players may block their opponent's path by occupying critical spaces. The game emphasizes spatial reasoning, foresight, and the balance between advancing one's own path and disrupting the opponent's progress.

## K. Insight Case Analysis

We analyze the high-quality insights stored in the memory bank across different games. These insights emerge from self-play trajectories and contribute to prompt optimization by encoding transferable strategic knowledge. We identify two primary categories of insights that improve game performance: (1) game-specific strategic principles that capture tactical knowledge, and (2) opponent modeling insights that focus on understanding and responding to other players.

**Game-Specific Strategic Principles.** These insights capture tactical knowledge that helps agents make better in-game decisions. They encode domain-specific heuristics that would otherwise require many episodes to rediscover.

---

**Kuhn Poker Strategic Insights**

**Insight 1 (Pressure-based betting):** "In future games, consider a strategy where you bet or call more frequently in early rounds, even with weaker cards, to increase potential pots and apply pressure on the opponent, especially when no initial bets are made."

**Insight 2 (Hand strength exploitation):** "In future games, players should adopt a more aggressive betting strategy when holding stronger cards, such as K, to force opponents into tough situations that might lead to folds or allow the player to take control of the pot more effectively."

---

*Figure 22.* Kuhn Poker strategic insights. These insights encode betting principles that balance aggression with hand strength, helping agents avoid predictable play patterns while maximizing expected value.

> **Briscola Strategic Insights**
>
> **Insight 1 (Trump timing):** "In future turns, prioritize using the Ace or trump cards at key moments to control the trick."
> **Insight 2 (Point maximization):** "When holding a trump card, prioritize using it to capture high-point non-trump cards led by the opponent, especially during the mid-game when more point cards are likely to be in play. This maximizes point gain and helps secure early leads."

*Figure 23.* Briscola strategic insights. These insights capture the timing and resource allocation principles for trump cards, enabling agents to maximize point capture rather than using high-value cards indiscriminately.

Strategic principles reduce the search space for decision-making by providing domain-appropriate heuristics. Rather than exploring all possible actions uniformly, agents can prioritize moves that align with proven tactical patterns, leading to faster convergence and more consistent performance.

**Opponent Modeling and Negotiation Dynamics.** These insights focus on understanding opponent behavior and leveraging psychological or structural aspects of multi-agent interactions.

> **Simple Negotiation Insights**
>
> **Insight 1 (Preference inference):** "To improve negotiation outcomes, Player should analyze the resource preferences of other Player more closely and tailor offers to match those preferences, possibly by proposing trades that highlight the mutual benefits rather than assuming equal value among resources."
> **Insight 2 (Information gathering):** "In future negotiations, it would be beneficial to engage in dialogue to better understand Player 1's resource priorities before making trade offers, potentially increasing the chance of acceptance and maximizing resource value."

*Figure 24.* Simple Negotiation insights. These insights reveal that players have asymmetric resource valuations, a concept not explicitly stated in the game description, and encourage proactive information gathering before committing to offers.

> **Two Dollars Negotiation Insight**
>
> **Insight (Time pressure leverage):** "To improve future negotiations, I could clearly convey the importance of reaching an agreement within the limited rounds available, perhaps framing my offer as a time-sensitive opportunity that other player wouldn't want to miss, thereby encouraging a quicker consensus."

*Figure 25.* Two Dollars insight. This insight captures a negotiation tactic that exploits the finite round structure, encouraging agents to use time pressure as a persuasion mechanism.

Opponent modeling insights enable agents to move beyond self-centered optimization toward strategic reasoning that accounts for the other player's objectives and constraints. By understanding that opponents have different preferences or that structural features like round limits can be leveraged, agents can craft more effective proposals and responses. These insights are particularly valuable in negotiation games where success depends on predicting and influencing opponent behavior.

## L. Prompt Case Analysis

Below, we present two optimized prompts in Figure 26 and Figure 27.

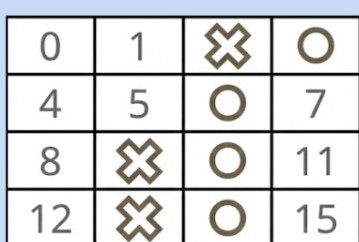

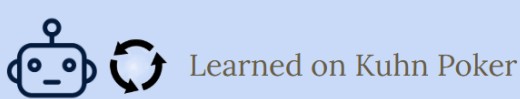

Base Prompt: You are playing a two-player zero-sum game. Make valid moves to win.submit the move enclosed by \\boxed{{}}. + [Game instruction]

Model Response: To create a path towards winning, I will place my stone in cell [5]. This move allows me to start building a connection towards the right side of the board.\n\n\\[\n\\boxed{5}\n\\]

Prompt learned from Self play: Incorporate a **bold betting** strategy when holding high cards (Q or K), using [Bet] to force **your opponent** into challenging decisions..... Remember to carefully review your options each round and ensure all moves are **formatted correctly**.+ [Game instruction]

Model Response: ...To apply a bold strategy and increase pressure on my opponent, I will place my stone in cell 7, which is a strategic move to block my opponent.\n\\boxed{[7]}

*Figure 26.* In this example (Fig. 26), the agent plays ×, while the opponent (○) is one move away from victory. The prompt on the right is learned via self-play in KUHN POKER and distills transferable behaviors, including opponent modeling, general strategic principles, and strict output-format constraints. Conditioned on this prompt, the agent handles the scenario more reliably; in this short-horizon case, the adapted prompt enables the agent to identify the correct blocking move and prevent the opponent's immediate win.

---

**Simple Negotiation System Prompt Transfer to Simple Tak Example**

Updated System Prompt:
" Evaluate Opponent current inventory and preferences before crafting an initial trade proposal. Offer a carefully curated mix of both lower-value and higher-value items that not only maximize your immediate gain but also align with Player 1's needs, enhancing the perceived mutual benefits of the exchange. Ensure your strategy incorporates diversified resources to attract Opponent interest, leveraging aggressive initial offers that target their most valued items while securing optimal benefits for yourself"
Sample Action:
"To maximize my chances of winning while also considering the current state of the board, I will place my stone in cell 6. This move not only blocks Player 0's potential path but also positions me closer to forming my own continuous path.\n\nThus, my move is: (boxed[6])"
"To maximize my chances of forming a continuous path while also considering the opponent's move, I will place my stone in cell 0. This move not only allows me to start building a path from the top edge but also blocks potential paths for Player 0.\n\nI will submit my move as follows: boxed[0]"
"he board is 4x4, and I have two adjacent O's in the top row (positions 4 and 5). The opponent has placed X's in 6 and 7, blocking the top row to the right. To build towards a left-right connection, I should extend my horizontal line leftward by placing in position 0, which is empty and adjacent to my O in 4. boxed[0]"

*Figure 27.* Simple Negotiation Prompt Transfer to Simple Tak. Although the updated prompt was not trained on Simple Tak, it encourages the model to explicitly reason from the opponent's perspective during its thought process, resulting in more consistent and reliable performance compared to the basic starting prompt.

## M. Algorithm Details

Algorithm 1 presents the full MEMO optimization loop, and Algorithm 2 details the replay-augmented tournament procedure invoked at each generation. All notation follows the main text (Sec. 2–3).

---

**Algorithm 1** MEMO: Memory-Augmented Context Optimization

---

**Require:** Base context $c_{\text{base}}$, optimizer LLM A, game environment $G$, population size $N$, generations $T$, proposal ratios $(r_{\text{rand}}, r_{\text{mem}})$ with $r_{\text{rand}} + r_{\text{mem}} = 1$, memory fraction $\pi$, TrueSkill penalty $\kappa$

**Ensure:** Optimized context $c^\star$

1:  $\mathcal{P} \leftarrow \{c_{\text{base}}\} \cup \{\text{RANDOMPROPOSAL}(A, c_{\text{base}})\}_{i=1}^{N-1}$                  ▷ Initialize candidate pool
2:  $\mathcal{C}_0 \leftarrow \text{TOPN}(\mathcal{P}, N)$                             ▷ Initial population
3:  $\mathcal{B}_{\text{mem}} \leftarrow \varnothing$                                ▷ Persistent memory bank
4:  $\mathcal{B}_{\text{rep}} \leftarrow \varnothing$                                    ▷ Replay buffer
5:  **for** $g = 0$ **to** $T-1$ **do**
6:       *// — Self-play tournament —*
7:       Inject memory subset $M \subseteq \mathcal{B}_{\text{mem}}$ into fraction $\pi$ of contexts in $\mathcal{C}_g$
8:       $\mathcal{R}_g \leftarrow \text{TOURNAMENT}(\mathcal{C}_g, G, \mathcal{B}_{\text{rep}})$                ▷ Play games (Alg. 2)
9:       Update TrueSkill ratings $(\mu_c, \sigma_c)$ from $\mathcal{R}_g$;   score $S(c) \leftarrow \mu_c - \kappa \sigma_c$
10:      *// — Trajectory reflection and memory update —*
11:      $\mathcal{W}_g \leftarrow \text{REFLECT}(A, \mathcal{R}_g)$           ▷ Extract typed insights from trajectories
12:      $\mathcal{B}_{\text{mem}} \leftarrow \text{CRUDUPDATE}(\mathcal{B}_{\text{mem}}, \mathcal{W}_g)$             ▷ Add / Edit / Remove
13:      *// — Context evolution —*
14:      $\mathcal{P} \leftarrow \text{RETAINTOP}(\mathcal{P} \cup \mathcal{C}_g)$           ▷ Update persistent candidate pool
15:      $n_r \leftarrow \lfloor N \cdot r_{\text{rand}} \rfloor$;    $n_m \leftarrow N - n_r$
16:      $\mathcal{U}_{\text{rand}} \leftarrow \text{RANDOMPROPOSAL}(A, \mathcal{P}, n_r)$           ▷ Style-guided edits
17:      $\mathcal{U}_{\text{mem}} \leftarrow \text{MEMORYPROPOSAL}(A, \mathcal{P}, \mathcal{B}_{\text{mem}}, n_m)$      ▷ Memory-informed edits
18:      $\mathcal{C}_{g+1} \leftarrow \text{TOPN}(\mathcal{P} \cup \mathcal{U}_{\text{rand}} \cup \mathcal{U}_{\text{mem}}, N)$
19: **end for**
20: **return** $c^\star = \arg\max_{c \in \mathcal{P}} S(c)$

---

**Algorithm 2** Replay-Augmented Tournament (called at line 8 of Alg. 1)

---

**Require:** Context population $\mathcal{C}_g$, game environment $G$, replay buffer $\mathcal{B}_{\text{rep}}$, priority exponent $\alpha$, replay probability $\beta$

**Ensure:** Trajectory set $\mathcal{R}_g$

1:  $\mathcal{R}_g \leftarrow \varnothing$
2:  **for** each scheduled game in the tournament **do**
3:       Sample $u \sim \mathcal{U}(0,1)$
4:       **if** $u < \beta$ and $|\mathcal{B}_{\text{rep}}| > 0$ **then**                    ▷ Replay from buffer
5:           Sample prefix $\tau_{\text{pre}}$ from $\mathcal{B}_{\text{rep}}$ with probability $p_i \propto \text{priority}(\tau_i)^\alpha$
6:           $\tau \leftarrow \text{PLAYFROMPREFIX}(G, \tau_{\text{pre}}, \mathcal{C}_g)$          ▷ Resume from stored state
7:       **else**                                            ▷ Fresh game
8:           $\tau \leftarrow \text{PLAYFRESH}(G, \mathcal{C}_g)$
9:       **end if**
10:      $\mathcal{R}_g \leftarrow \mathcal{R}_g \cup \{\tau\}$
11:      $\mathcal{B}_{\text{rep}} \leftarrow \text{INSERT}(\mathcal{B}_{\text{rep}}, \tau)$           ▷ Store with inverse-frequency priority
12: **end for**
13: **return** $\mathcal{R}_g$

---

**Notation Summary.** $\mathcal{C}_g$: context population at generation $g$; $\mathcal{P}$: persistent candidate pool storing the best contexts across all generations; $\mathcal{B}_{\text{mem}}$: memory bank accumulating structured insights; $\mathcal{B}_{\text{rep}}$: replay buffer storing trajectory prefixes with environment seeds; $S(c) = \mu_c - \kappa\sigma_c$: TrueSkill lower-confidence score (Eq. 1); $\pi$: fraction of population receiving memory at inference time; $\text{priority}(\tau) = 1/\text{count}(\tau)$: inverse-frequency priority for replay sampling.

## N. Additional Ablation Studies

This appendix presents additional ablations that vary axes of the experimental setting (game horizon, number of players, reward structure, and self-play protocol) while holding the rest of the MEMO pipeline fixed. Each ablation isolates one

experimental factor and reports its effect on win rate relative to the default setting in Sec. 4: the horizon (Sec. N.1), the number of players (Sec. N.2), the reward structure (Sec. N.3), and the self-play protocol (Sec. N.4).

### N.1. Longer-Horizon Games

To test whether MEMO's gains depend on the horizon of the game, we increase SIMPLETAK from a $4 \times 4$ board to a $10 \times 10$ board while keeping the optimization pipeline fixed. The board change raises the average episode length from roughly 12 to 62 turns. After optimizing GPT-4o-mini with 2,000 self-play games, we evaluate against two held-out opponents over 50 games each.

*Table 15.* Longer-horizon SIMPLETAK ablation with GPT-4o-mini. Board size is increased from $4 \times 4$ to $10 \times 10$, raising average game length from $\sim 12$ to $\sim 62$ turns. Results are win rates over 50 games against each held-out opponent.

| Opponent | Baseline | MEMO | $\Delta$ |
|---|---|---|---|
| Gemini-2.5-Flash-Lite | 54.0% | 64.0% | +10.0% |
| Grok-4-Fast-Non-Reasoning | 46.0% | 58.0% | +12.0% |
| Mean | 50.0% | 61.0% | +11.0% |

MEMO raises mean win rate from 50.0% to 61.0% on the longer board, with consistent gains against both opponents. The persistent memory bank continues to compound useful insights even when the trajectory length grows by roughly $5\times$, indicating that the gains in Sec. 4 are not an artifact of the short-horizon default board.

### N.2. Three-Player Games

We extend MEMO to a three-player setting using `ThreePlayerIPD-v0`, a three-player iterated prisoner's dilemma with free-chat communication and coalition dynamics. Tournament selection, the persistent memory bank, and prioritized replay all operate over completed trajectories and make no assumption about the number of players, so the same pipeline applies without modification. We optimize GPT-4o-mini with 2,000 self-play games and evaluate against two held-out opponents.

*Table 16.* Three-player IPD ablation with GPT-4o-mini. Results are win rates against each held-out opponent.

| Method | vs Gemini | vs Grok | Average |
|---|---|---|---|
| Baseline | 8.0% | 21.3% | 14.7% |
| CoT | 10.0% | 23.3% | 16.7% |
| MEMO | **25.3%** | **32.0%** | **28.7%** |

MEMO nearly doubles the baseline average win rate, from 14.7% to 28.7%, and also outperforms the CoT baseline at 17.8%. Because the three-player setting introduces coalition formation and potentially non-transitive interactions, this ablation indicates that MEMO's gains are not restricted to two-player zero-sum games.

### N.3. Non-Zero-Sum Games: MultiAgentBench Bargaining

The five default tasks in Sec. 4 are two-player zero-sum. We move to a non-zero-sum reward structure, where optimal play is no longer purely adversarial, by evaluating on the Bargaining task from MultiAgentBench (Zhu et al., 2025). Bargaining is non-zero-sum because both buyer and seller gain from closing a deal at a mutually acceptable price, so the optimum is a negotiated agreement rather than the buyer's adversarial best response. We optimize a GPT-4o-mini buyer with MEMO and evaluate it against three held-out sellers that span recent capable models: Gemini-2.5-Flash-Lite, Qwen3-235B, and Grok-4-Fast.

*Table 17.* Non-zero-sum ablation on the MultiAgentBench Bargaining task. Buyer is GPT-4o-mini optimized with MEMO; sellers are held-out frontier models. Deal Rate is the fraction of negotiations that close, Savings (%) is the buyer's price reduction relative to the seller's anchor, and Buyer Score is the LLM-judge rating of buyer behavior. Best per seller block in **bold**.

| Seller | Method | Deal Rate | Savings (%) | Buyer Score |
|---|---|---|---|---|
| Gemini-2.5-Flash-Lite | Baseline | 73% | 17.5 | 3.67 |
| | CoT | 83% | 18.8 | 3.71 |
| | MEMO | **90%** | **19.4** | **3.80** |
| Qwen3-235B | Baseline | 80% | 23.1 | 3.50 |
| | CoT | **87%** | 23.9 | **3.79** |
| | MEMO | 80% | **26.7** | **3.79** |
| Grok-4-Fast | Baseline | 30% | 22.9 | 3.45 |
| | CoT | 10% | 27.2 | 3.39 |
| | MEMO | **60%** | **34.7** | **3.51** |

Averaged across the three sellers, MEMO improves the deal rate from 61% to 77%, savings from 21.2% to 26.9%, and buyer score from 3.54 to 3.70. The improvement holds even against the strongest seller, Grok-4-Fast, where the baseline closes only 30% of negotiations while MEMO closes 60%. The persistent insights distilled by MEMO transfer from zero-sum tournaments to mixed-motive negotiation. Success in this setting requires balancing aggressive offers with concessions that keep the counterparty at the table.

### N.4. Self-Play Variant: One-Sided vs. Round-Robin

MEMO's default self-play protocol is *one-sided context optimization*. Only the focal player's context is optimized, while the opponent uses a fixed baseline context. We compare this default against a *round-robin* variant in which both players' contexts co-evolve. Both variants are run on KuhnPoker with GPT-4o-mini for three independent trials, and we report each trial's win rate against held-out opponents.

*Table 18.* Self-play variant ablation on KuhnPoker with GPT-4o-mini. Format: per-trial win rate, mean (RSE) across three trials.

| Variant | Trial 1 | Trial 2 | Trial 3 | Mean (RSE) |
|---|---|---|---|---|
| One-sided (default) | 54.0% | 57.3% | 55.3% | 55.6% (1.7%) |
| Round-robin | 55.6% | 58.6% | 50.3% | 54.8% (4.4%) |

The two variants reach comparable mean win rates (55.6% vs. 54.8%), but the round-robin setup exhibits roughly 2.4× higher relative standard error (4.4% vs. 1.7%). A co-evolving opponent injects additional uncertainty into the tournament signal, which destabilizes selection across runs without improving mean performance. We therefore retain the one-sided variant as the default, since it yields lower run-to-run dispersion at no cost in win rate.

GEPA scoring standard

# invalid action
score = 0.0
feedback = "Your predicted action is invalid. Please ensure that your action is a valid move in the game. Here is the reasoning process {{model_raw_output}}. Think about how you could have reasoned to choose a valid action that leads to a WIN."

# Win Trajectory
# Action match
score = 1.0
feedback = "You correctly predicted the action {{pred_action}} that led to a WIN. This action was indeed the one taken in the winning trajectory. Great job!"

# Action mismatch
score = 0.0
feedback = "You predicted the action {{pred_action}}, but the action taken in the winning trajectory was {{traj_action}}. This mismatch means you did not predict the winning action correctly. Here is the reasoning process {{pred_raw_action}}. Think about how you could have reasoned to get the correct action."

# Lose Trajectory
# Action match
score = 0.0
feedback = "You correctly predicted the action {{pred_action}} that led to a LOSE. However, this action was part of a losing trajectory. While your prediction matches the trajectory, it did not lead to a win. Here is the reasoning process {{pred_raw_action}}. Think about how you could have reasoned to choose an action that leads to a WIN."

# Action mismatch
score = 1.0
feedback = "You predicted the action {{pred_action}}, but the action taken in the losing trajectory was {{traj_action}}. This mismatch means you did not predict the losing action correctly. Here is the reasoning process {{pred_raw_action}}. Think about how you could have reasoned to choose an action that leads to a WIN."

# Draw Trajectory
# Action match
score = 0.0
feedback = "You predicted the action {{pred_action}}, which matches the action taken in the TIE trajectory. However, since the trajectory resulted in a TIE, this not help in achieving a WIN. Here is the reasoning process {{pred_raw_action}}. Think about how you could have reasoned to choose an action that leads to a WIN."

# Action mismatch
score = 1.0
feedback = "You predicted the action {{pred_action}}, but the action taken in the TIE trajectory was {{traj_action}}. This mismatch means you did not predict the TIE action correctly. Here is the reasoning process {{pred_raw_action}}. Think about how you could have reasoned to choose an action that leads to a WIN."

*Figure 21.* GEPA scoring standard

