# OpenReview forum: "MEMO: Memory-Augmented Model Context Optimization for Robust Multi-Turn Multi-Agent LLM Games"
_ICML.cc/2026/Conference — ICML 2026 regular_

### Official Review · Reviewer_Rbw3 · 2026-03-12

**Soundness:** 3
**Presentation:** 4
**Significance:** 4
**Originality:** 3
**Overall Recommendation:** 5
**Confidence:** 3

**Summary:**

The paper proposes MEMO, a weight-free self-play framework for multi-turn, multi-agent LLM game settings that optimizes inference-time context rather than model weights. MEMO couples exploration with retention to reduce run-to-run variance and improve win rates. Across five text-based games and two base models, MEMO reports substantial gains over static prompting and prompt-optimization baselines at a modest interaction budget, competitive performance with RL on some tasks, and markedly reduced run-to-run dispersion.

**Compliance With Llm Reviewing Policy:**

Affirmed.

**Final Justification:**

My concerns have been adequately addressed. I am keeping my current score.

**Key Questions For Authors:**

* How did you measure the "small early deviations" mentioned in the abstract and intro? And how do you prove your method actually prevents their error accumulation?
* Why is the evaluation temperature fixed at 1.0 throughout?
* How do you ensure the LLM-driven CRUD operations for the memory bank don’t lead to low-quality or hallucinated content, since no quality safeguards are mentioned?

**Limitations:**

yes

**Strengths And Weaknesses:**

# Strengths
* Addresses an increasingly important problem: instability and sensitivity in multi-turn, multi-agent LLM evaluations.
* Transparent loop design with population size, number of generations, and game budgets; easy-to-follow schematic of the pipeline.
* Thoughtful analyses on prompt-induced ranking instability and cross-game/model transfer.

# Weaknesses
* The evaluation temperature is set to 1.0 throughout, which can artificially inflate variance; no sensitivity to decoding settings is reported.
* Token accounting reports output tokens only; total cost (input + output) and memory-context length overhead are not provided, making cost-effectiveness comparisons incomplete.
* The method evaluates over three sets of 50 games against the **same opponent**, which may lead to overfitting to that specific opponent rather than generalizable, robust policy improvement.
* The memory bank relies on LLM‑driven CRUD operations, but the paper lacks explicit quality safeguards.
* In Table 1, Mean Win Rate, both UnstableBaseline and MEMO are highlighted in bold, which is confusing and ambiguous.

---

> ### Author Rebuttal · Authors · 2026-03-31
>
> We thank the reviewer for the thorough evaluation.
>
> > **W1 / Q2**: The evaluation temperature is set to 1.0 throughout, which can artificially inflate variance ... Why is the evaluation temperature fixed at 1.0?
>
> We conducted additional experiments on KuhnPoker with GPT-4o-mini, sweeping the sampling temperature over $\tau \in \{0.1, 0.5, 1.0\}$. Each configuration was tested against three held-out opponents. We report mean win rates with RSE in parentheses.
>
> | **Method** | **$\tau=0.1$** | **$\tau=0.5$** | **$\tau=1.0$** |
> |---|---:|---:|---:|
> | Baseline | 43.2% (5.2%) | 37.2% (8.5%) | 33.2% (0.5%) |
> | CoT | 44.3% (2.1%) | 38.1% (2.5%) | 44.0% (4.3%) |
> | MEMO | 54.9% (6.7%) | 56.4% (1.4%) | 52.4% (1.7%) |
>
> RSE shows no monotonic relationship with temperature. The highest RSE for each method occurs at a different $\tau$, indicating that variance depends on the interaction between both players' sampling behavior, not temperature alone. MEMO outperforms both baselines at every temperature, confirming gains are not an artifact of high-temperature sampling. We use $\tau=1.0$ because it is the default for most LLM APIs.
>
> ---
>
> > **W2**: Token accounting reports output tokens only; total cost ... and memory-context length overhead are not provided ...
>
> |  | **MEMO** | **TextGrad** | **MIPRO** | **GEPA** |
> |------------------------|---------:|-------------:|----------:|---------:|
> | **overall cost (optimizer + self play tournament)** | $3.42 | $3.02 | $4.64 | $3.46 |
> | **optimizer only cost** | $0.11 | $0.12 | $0.55 | $0.47 |
>
> On SimpleNegotiation, MEMO's total cost is comparable to TextGrad and cheaper than MIPRO and GEPA, confirming gains are not explained by higher spending.
>
> Prompt optimization methods (TextGrad, MIPRO, GEPA) do not use memory at inference, so memory-context overhead does not apply to them. Among memory-based methods, MEMO injects 948 tokens per turn, compared to ~1,004 for Reflexion (Shinn et al., NeurIPS 2023) and ~801 for mem0 (Chhikara et al., 2025). MEMO achieves 54.9% win rate versus 38.7% (Reflexion) and 37.1% (mem0), showing that it uses less memory than Reflexion while outperforming both.
>
> ---
>
> > **W3**: The method evaluates over three sets of 50 games against the same opponent, which may lead to overfitting ... rather than generalizable, robust policy improvement.
>
> To mitigate overfitting, our training phase uses self-play with different starting prompts. We subsequently evaluated each method against **three diverse opponents**: Gemini 2.5 Flash Lite, Grok 4 Fast Non-Reasoning, and Qwen 3-235B. The learned insights and state abstracts from self-play were reused across all three opponents rather than being optimized separately for each one, confirming generalizability.
>
> ---
>
> > **W4 / Q3**: The memory bank relies on LLM-driven CRUD operations, but ... lacks explicit quality safeguards. ... How do you ensure ... CRUD operations don't lead to low-quality or hallucinated content?
>
> We agree this deserves more discussion. Contexts with low-quality or hallucinated memory entries lose games and are discarded by TrueSkill selection. Table 3 confirms the net effect is positive across all games. Our Insight Case Analysis in the appendix confirms that retained insights are coherent and strategically meaningful. We acknowledge that explicit safeguards are a promising direction for future work.
>
> ---
>
> > **W5**: In Table 1 ... both UnstableBaseline and MEMO are highlighted in bold, which is confusing ...
>
> Thank you for the observation. We will revise Table 1 so that only the UnstableBaseline result is highlighted in bold.
>
> ---
>
> > **Q1**: How did you measure the "small early deviations" ...? And how do you prove your method actually prevents their error accumulation?
>
> We measure the aggregate effect of small deviations rather than individual turn-level divergences. Our prompt sensitivity study (Appendix A, Figure 4) shows that five semantically equivalent KuhnPoker prompts produce Kendall $\tau_b$ ranging from $-0.504$ to $+0.500$, indicating that minor prompt variations alone flip pairwise model rankings. This confirms that small contextual differences cascade into large outcome shifts, consistent with prior works (Laban et al., 2025; Cemri et al., 2025). The baseline RSE of 44.9% also quantifies the resulting instability. MEMO addresses this at the context level. Three mechanisms contribute: (1) TrueSkill uncertainty-aware selection avoids promoting contexts that appear strong by luck in small samples, (2) the persistent memory bank anchors independent runs toward shared strategic insights rather than letting them drift apart, and (3) multi-game evaluation averages out single-trajectory noise. Together, these reduce RSE from 44.9% to 6.4%.

---

> > ### Author Rebuttal · Reviewer_Rbw3 · 2026-04-02
> >
> > Thanks for the detailed rebuttal.

---

> > > ### Author Response · Authors · 2026-04-03
> > >
> > > Thank you for the positive acknowledgement! We are glad that our rebuttal was able to address your concerns.

---

### Official Review · Reviewer_aUo5 · 2026-03-13

**Soundness:** 3
**Presentation:** 3
**Significance:** 3
**Originality:** 2
**Overall Recommendation:** 4
**Confidence:** 4

**Summary:**

The paper proposes MEMO to address the instability and underperformance in interactive games with LLM agents. MEMO is a self-play framework that treats inference-time context as an optimizable object by coupling retention and exploration. A memory bank containing distilled trajectories is used for getting insights and used for priors in the subsequent play. Prompt evolution with uncertainty-aware selection in a tournament-like structure is combined with prioritized replay to revisit vital states for sample efficient coverage. MEMO obtains substantial performance gain for GPT-4o-mini and Qwen-2.5-7B-Instruct using 2000 self play games per task.

**Compliance With Llm Reviewing Policy:**

Affirmed.

**Final Justification:**

The authors have provided additional results that effectively address my concern raised in the review. I requested for further detail about the experiment design, which was also explained well by the authors' following response.

**Key Questions For Authors:**

## Questions
1. Could you report the entire API cost and wall clock time of running MEMO and other algorithms that are compared? What would happen if the methods get the same amount of API credit?
2. Could you compare the memory consumption of MEMO and other algorithms?
3. Are the contexts all considered new after random proposals or memory-augmented updates? then are the TrueSkill ratings of contexts all reset in the next generation as well?
4. Is the priority metric effective when the trajectories are hard to be compared and "counted"? (ex. multi-agent debate in natural language)

## Suggestions
1. Table 1: It seems in Briscola, ToT is performing better than MEMO (GPT-4o-mini). Both the MEMO and UnstableBaseline's performances are bolded in Qwen2.5-7B-Instruct results in Mean Win Rate.
2. Some texts in the figures can be made larger to improve visibility, while the figures themselves do not have to be larger than now.

**Limitations:**

I think a bit more discussion of the possible limitations of MEMO can be helpful for the readers (such as general limitation of prompt optimization methods leaving the modification of the prompts entirely to LLMs).

**Strengths And Weaknesses:**

## Strengths
1. The approach is well motivated and the experiment results well demonstrate the effectiveness of the method.
2. The experiments are conducted extensively, performing multiple runs with the same setting as well.
3. The paper is well organized in general, explaining most of the relevant concepts within the main text.
4. The proposed method is analyzed from several different perspectives, demonstrating the effectiveness of it.

## Weaknesses
1. The paper seems to be missing some crucial information about MEMO, such as API cost, wall clock time for running the algorithm and memory consumption.
2. While effective, the paper does not provide rigorous theoretical analysis of the proposed method.

---

> ### Author Rebuttal · Authors · 2026-03-31
>
> > **W1/Q1**:
> >
> > ... missing some crucial information about MEMO, such as API cost ... Could you report the entire API cost ... of running MEMO and other algorithms ...?
>
> Cost for all methods on SimpleNegotiation, GPT-4o-mini, 2,000 self-play games:
>
> |  | **MEMO** | **TextGrad** | **MIPRO** | **GEPA** |
> |---|---:|---:|---:|---:|
> | **Overall cost** | $3.42 | $3.02 | $4.64 | $3.46 |
> | **Optimizer only** | $0.11 | $0.12 | $0.55 | $0.47 |
>
> Overall cost includes the optimizer and self-play tournament; optimizer only reflects prompt evolution alone. MEMO's $3.42 overall cost is the second lowest with the best performance.
>
> > What would happen if the methods get the same amount of API credit?
>
> MEMO's gains are not from higher spending. With equal budget, MEMO could run more optimization rounds, potentially improving performance further.
>
> > ... wall clock time for running the algorithm ...
>
> All methods rely on external API calls whose latency fluctuates with server load, rate limits, and concurrency settings, so a fair wall-clock comparison across methods is not feasible under these conditions.
>
> > ... memory consumption.
>
> The other methods do not maintain a memory bank. MEMO achieves the highest win rate and lowest variance with moderate memory consumption; see Q2 for details.
>
> ---
>
> > **W2**: While effective, the paper does not provide rigorous theoretical analysis of the proposed method.
>
> MEMO does not provide a formal theoretical guarantee, consistent with existing methods. Extending the framework with theoretical analysis of convergence or memory quality is an interesting direction.
>
> ---
>
> > **Q2**: Could you compare the memory consumption of MEMO and other algorithms?
>
> The baselines in our paper, TextGrad, MIPRO, GEPA, UnstableBaseline, and SPIRAL, do not maintain memory banks. We evaluated Reflexion (Shinn et al., NeurIPS 2023) and mem0 (Chhikara et al., 2025) on SimpleNegotiation with GPT-4o-mini, 2,000 training games, and 3 runs $\times$ 50 games per opponent.
>
> | Method | Mean Win Rate | RSE | Bank (tokens) | Eval (tokens) |
> |---|---|---|---|---|
> | Baseline | 34.7% | 14.0% | 0 | 0 |
> | Reflexion | 38.7% | 9.2% | 98,394 | 1,004 |
> | mem0 | 37.1% | 11.9% | **2,443** | **801** |
> | **MEMO (ours)** | **54.9%** | **6.4%** | 15,065 | 948 |
>
> **Bank** = total memory from training. **Eval** = tokens injected at evaluation.
>
> MEMO achieves the highest win rate at 54.9%, outperforming Reflexion at 38.7% and mem0 at 37.1%, with the lowest RSE at 6.4%. Reflexion accumulates 98K tokens but only the last 20 fit at evaluation, discarding most experience. mem0 extracts memories from individual message pairs rather than full trajectories, missing multi-turn lessons. MEMO's CRUD operations consolidate all 2,000 games into 15K tokens of refined insights, retaining lessons both methods lose.
>
> ---
>
> > **Q3**: Are the contexts all considered new after random proposals or memory-augmented updates? ... are the TrueSkill ratings ... all reset in the next generation as well?
>
> Yes. MEMO maintains a candidate pool of agents, each with a different context and TrueSkill rating. Each generation introduces new or mutated agents, changing the pool's composition. Because the pool differs across generations, all ratings are reset, forcing every agent, including retained elites, to re-prove itself.
>
> ---
>
> > **Q4**: Is the priority metric effective when the trajectories are hard to be compared and "counted"?
>
> Our default priority metric relies on structured action sequences, e.g., `[Offer: 3 Sheep -> 5 Brick]`, for state-visitation counting, which cannot apply to free-form natural language. We addressed this with an LLM-based state abstraction that summarizes each game state into an abstract category, e.g., "aggressive opening offer" or "concession after rejection." We validated this on MultiAgentBench Bargaining (Zhu et al., ACL 2025) with GPT-4o-mini, 2,000 self-play games, and three sellers.
>
> | Seller | Method | Deal Rate | Savings (%) | Buyer Score |
> |---|---|---|---|---|
> | Gemini-2.5-Flash-Lite | Baseline | 73% | 17.5 | 3.67 |
> | | CoT | 83% | 18.8 | 3.71 |
> | | **MEMO** | **90%** | **19.4** | **3.80** |
> | Qwen3-235B | Baseline | 80% | 23.1 | 3.50 |
> | | CoT | 87% | 23.9 | **3.79** |
> | | **MEMO** | 80% | **26.7** | **3.79** |
> | Grok-4-Fast | Baseline | 30% | 22.9 | 3.45 |
> | | CoT | 10% | 27.2 | 3.39 |
> | | **MEMO** | **60%** | **34.7** | **3.51** |
>
> MEMO improves average deal rate from 61% to 77%, savings from 21.2% to 26.9%, and buyer score from 3.54 to 3.70, confirming that the priority metric remains effective in natural language settings.
>
> ---
>
> > **S1,S2**: ... Both the MEMO and UnstableBaseline's performances are bolded ... in Mean Win Rate. ... Some texts in the figures can be made larger to improve visibility ...
>
> We will bold only the best method per column in Table 1 in our manuscript, add a note on Briscola favoring tree-search, and increase figure font sizes.

---

> > ### Author Rebuttal · Reviewer_aUo5 · 2026-04-01
> >
> > Thank you for the additional results and clarifications.
> >
> > > We addressed this with an LLM-based state abstraction that summarizes each game state into an abstract category, e.g., "aggressive opening offer" or "concession after rejection."
> >
> > I find this strategy interesting. Could you explain the details of how the state abstraction is performed? What prompt did you use? Does this procedure require referring to the description of the environment, or does it summarize the states after seeing sample episodes? Is this process automated, so that the user does not have to provide information to the LLM manually?
> >
> > ### Tiny Suggestion
> > It seems there is slight difference between how the author responses are written, depending on the reviewer. While this is a minor thing and apart from the technical details of the paper, I think maintaining a consistent and respectful tone across reviewers is a safer strategy.

---

> > > ### Author Response · Authors · 2026-04-03
> > >
> > > We thank the reviewer for the constructive follow-up and for the helpful suggestion on tone. We appreciate you pointing this out. The inconsistency was not intentional. Some responses were necessarily compressed by the rebuttal character limit, and we will be more careful going forward.
> > >
> > > The LLM-based state abstraction is a fully automated training-time step used only for prioritized replay in free-form language settings such as Bargaining in MultiAgentBench. Its purpose is to group similar game situations together so that trajectories passing through underrepresented states are sampled more often during training.
> > >
> > > At each turn, we call the same GPT-4o-mini model used for gameplay to label the current state. The input includes the environment/task description with rules and any role or private instructions, the current round/turn information, and the dialogue/offer history so far. The model outputs one short strategic label for the current partial state, such as “aggressive opening offer,” “concession after rejection,” “deadline pressure,” or “fairness-based appeal.” We use that label to group similar game situations, so that trajectories passing through underrepresented states are sampled more often during training.
> > >
> > > The prompt follows this template:
> > >
> > > > *”Given the game description and the interaction so far, summarize the current \<Game\_Name\> situation into one short reusable strategic category.*
> > > >
> > > > *Existing abstract states:*
> > > > *[PASTE\_EXISTING\_ABSTRACT\_STATES\_HERE]*
> > > >
> > > > *Prefer an existing abstract-state label if one clearly fits; otherwise, propose a new label in the same style.*
> > > > *Focus on the stage of the game and the agent’s stance. Output only the category, not an explanation.”*
> > >
> > > The list of existing abstract states starts empty and accumulates labels as episodes are processed. When labeling a new turn, the model sees all labels generated so far and reuses an existing one if it fits, keeping the taxonomy compact. The examples mentioned above, such as “aggressive opening offer” and “concession after rejection,” are representative outputs of this prompt.
> > >
> > > We address each question below.
> > >
> > > (1) Yes, we do include the environment/task description, because labels such as “concession,” “fairness appeal,” or “deadline pressure” are task-dependent.
> > >
> > > (2) No, this does not require first reviewing a batch of sample episodes to manually induce a taxonomy. Instead, each partial episode is abstracted online from its current textual state. During replay, we first sample an abstract state and then sample a specific trajectory from that state.
> > >
> > > (3) Yes, the process is fully automated. The user does not need to provide labels or state annotations manually.
> > >
> > > This abstraction is used only to define replay counts in natural-language settings. It does not inject extra privileged information into the evaluation agent. Its role is simply to make replay grouping meaningful when exact string matching over free-form dialogue would be too brittle.
> > >
> > > We agree this detail should have been described more explicitly, and we will add the abstraction prompt and full automation details to the appendix in the revision.

---

### Official Review · Reviewer_rHEi · 2026-03-13

**Soundness:** 3
**Presentation:** 2
**Significance:** 3
**Originality:** 2
**Overall Recommendation:** 4
**Confidence:** 3

**Summary:**

The authors propose MEMO which combines: 1) evolutionary context search that optimizes for an objective that focuses on the reliability of goal achievement by disincentivizing that standard deviation of outcomes with 2) a persistent memory to add game-specific prior knowledge that prioritizes rare and informative states. MEMO is a weight-free self-play framework that tries to distill key insights from a persistent memory bank while exploring through tournament-style prompt evolution. MEMO showcases superior performance to relevant baselines using the same base model across 5 text based games and also show significant efficiency in comparison to the most competitive RL baselines. Ablation studies also demonstrate that each major component of the system adds value.

**Compliance With Llm Reviewing Policy:**

Affirmed.

**Key Questions For Authors:**

Q1: Is there evidence that your approach has directly addressed the run-to-run variance problem established in your motivation?

Q2: Can you comment on the class of settings / applications your approach is limited to?

Q3: Can you comment on the heuristic nature of TrueSkill and the value / risks of optimizing for something other the expected reward?

Q4: Can you comment on the heuristic nature of the way memories are prioritized and settings where this may be beneficial or harmful?

Q5: Can you please provide a brief overview of each of the "prompt" and "RL" baselines you consider highlighting their motivation and the key differences with your implementation of MEMO?

**Limitations:**

No. Many design choices are presented without providing insight about the potential limitations of these choices outside of a particular domain of problems. For example, 1) the setting assumes a game with binary outcome signals in which self-play is possible, 2) TrueSkill is a heuristic -- generally speaking it does not optimize for the expected reward due to the focus on reliability, 3) the prioritization of memories is also based on a heuristic, it can often be limited to remove memories of failure, which could also have implications for safety.

**Strengths And Weaknesses:**

Strengths:
- This work presents very strong results both in terms of performance and efficiency.
- They compare to a large set of baselines across a variety of categories.
- They perform key ablations validating the empirical value of the different core modules.
- They provide analysis of algorithmic efficiency and not just performance.


Weaknesses:
- The paper lacks strong theoretical grounding. The authors mention the run-to-run variance problem, but it was not clear to me exactly how and to what degree this is addressed by their approach. Moreover, many of the decisions that are introduced can clearly be seen as heuristics and there is not sufficient discussion of the limitations of these heuristics.
- It would be much better if the related work preceded the experiments section and there should really be a detailed breakdown of the baselines being compared against. There are a lot of superficial comparisons i.e. it is more efficient than X without really providing readers a clear picture of what X is an the context in which it was proposed.
- The overall novelty of the approach seems limited. It seems like no one has combined this whole pipeline before, but it seemed that each individual piece had pretty much been introduced in the prior literature or is heavily influenced by it.

---

> ### Author Rebuttal · Authors · 2026-03-31
>
> We thank the reviewer for recognizing the breadth of the experiments and for the detailed feedback. We address each concern below.
>
> ---
>
> **W1**:
> > The paper lacks strong theoretical grounding...
>
> MEMO does not provide a formal theoretical guarantee, consistent with existing prompt optimization methods which are similarly empirically motivated. Developing a theoretical framework for this setting is an important direction for future work.
>
>
> > **W1/Q1** ... it was not clear to me ... how and to what degree the run-to-run variance problem is addressed ...
>
>
> **How.** Different from standard prompt optimizers, MEMO's persistent memory accumulates reusable insights across generations, anchoring independent runs to a common set of distilled strategies.
>
> **To what degree.** We measure this via Relative Standard Error (RSE): on GPT-4o-mini, MEMO achieves 6.4% RSE versus 44.9% for baseline and 11.3% for GEPA, a 7$\times$ reduction. On Qwen2.5-7B-Instruct, 6.1% versus 30.1% and 14.8%.
>
>
> > ... many of the decisions ... can clearly be seen as heuristics ... not sufficient discussion of the limitations ...
>
> We discuss the two primary heuristics below: TrueSkill selection (W5/Q3) and LLM-driven CRUD operations (W6/Q4).
>
> ---
>
> > **W2/Q5**:
> >
> > ... the related work preceded the experiments section ... there should really be a detailed breakdown of the baselines ...
>
> > ... provide a brief overview of each ... baseline ... highlighting ... key differences with ... MEMO?
>
> Detailed baseline descriptions, adaptation procedures, and implementation specifics are provided in Appendix G and H in our manuscript. In the revision, we will move Related Work before Experiments and add a concise baseline overview table.
>
> ---
>
> > **W3**: The overall novelty of the approach seems limited. ... each individual piece had pretty much been introduced in the prior literature ...
>
> MEMO is not a simple combination. Prior work explored prompt optimization and memory independently; we diagnose why each alone fails: prompt optimization without memory is run-dependent, while memory without exploration accumulates unvalidated insights.
>
> **Prompt optimization without memory.** Methods such as TextGrad (Yuksekgonul et al., 2024), MIPRO (Opsahl-Ong et al., 2024), and GEPA (Agrawal et al., 2025) update prompts from local trajectory batches without retaining insights across rounds, so each run converges to a different local solution. MEMO's persistent memory fixes this by accumulating distilled strategies across generations. Table 2 in our manuscript confirms that MEMO outperforms all three with higher win rate and lower RSE.
>
> **Memory without exploration.** Methods like Reflexion (Shinn et al., NeurIPS 2023) and ExpeL (Zhao et al., AAAI 2024) accumulate memories but lack structured exploration to validate them against diverse opponents. Table 3 in our manuscript shows that memory alone gives +10.4% and exploration alone gives +3.3%, but combining them yields +24.3%, more than the sum of their parts.
>
> ---
>
> > **W5/Q3**: TrueSkill is a heuristic ... it does not optimize for the expected reward due to the focus on reliability. ... Can you comment on the value / risks ...?
>
> TrueSkill is indeed a heuristic that prioritizes reliability over raw win rate. Raw win rate from small samples is noisy, so TrueSkill penalizes uncertainty, requiring consistent performance before selection. As sample size grows, it converges to the true win rate. Table 1 in our manuscript confirms this: MEMO achieves the lowest RSE. The main risk is discarding strong but undertested contexts; we mitigate this via a persistent pool that carries top candidates forward across generations.
>
>
> ---
>
> > **W6/Q4**:
> >
> > ... the prioritization of memories is ... based on a heuristic ... removing memories of failure ... could have implications for safety.
>
> > ... Can you comment on ... settings where this may be beneficial or harmful?
>
> MEMO does **not** simply remove failure memories. The CRUD operations Add novel insights from both wins and losses, Remove contradictory pairs (conflict resolution, not failure suppression), and Edit similar insights into more conditional statements. LLM-driven judgments for these operations are inherently noisy, but we mitigate this by organizing insights into categorized fields. Despite this noise, memory operations consistently improve performance across all tested games, as shown in Table 3 in our manuscript.
>
> ---
>
> > **Q2**: ... Can you comment on the class of settings / applications your approach is limited to?
>
> MEMO applies to multi-turn, multi-agent interactions with clear outcome signals (win/loss/draw), covering games, negotiation, debate, and red-teaming. Our Multiagentbench Bargaining experiment (see Reviewer 5wpg W1/Q1) confirms generalization to mixed-motive settings. The main limitation is open-ended tasks without discrete outcomes; extending to continuous rewards is a promising direction.

---

> > ### Author Rebuttal · Reviewer_rHEi · 2026-04-01
> >
> > For W1, I am not asking for "a formal theoretical guarantee." I am more getting at the lack of a first principles based analysis and the embrace of heuristics.
> >
> > Thank you for your answer regarding the run to run variance.
> >
> > For W3, I am really confused about what you mean by the term "run" and found this response unconvincing in rebutting my point overall.
> >
> > For Q3, on what basis do you claim "as sample size grows, it converges to the true win rate". I feel like you didn't really engage with my request.
> >
> > For Q2, can you elaborate more on the dependence on the horizon? When you say discrete outcomes, do you simply mean a finite number of possible outcomes or is there a further restriction / limitation?

---

> > > ### Author Response · Authors · 2026-04-03
> > >
> > > We thank the reviewer for the thoughtful follow-up.
> > >
> > > > **W1**: ...lack of a first principles based analysis....
> > >
> > > The first-principles view we have in mind is the following. In this setting, optimization is noisy for two separate reasons. First, terminal outcomes are sparse and coupled across multiple turns. Second, standard prompt optimizers are generation-local, so useful but rare lessons discovered in one round are forgotten in the next. Two design requirements follow from this. Selection should be conservative under small samples, and useful strategic information should persist across generations rather than being repeatedly rediscovered.
> > >
> > > MEMO's components are practical instantiations of those principles, not claimed theoretically optimal rules. TrueSkill is used as a conservative small-sample selection rule, and persistent memory is used to reduce generation-locality by carrying forward distilled strategies. Without curation, stored insights grow to over 160 by generation 4, producing conflicting guidance. CRUD operations keep the count at 38.7. We agree the paper is empirical in how these instantiations are chosen, and we will foreground these motivations in the revision.
> > >
> > >
> > > ---
> > >
> > > > **W3**: ... confused about the term "run" ... found this response unconvincing ...
> > >
> > > On terminology, a "run" is one complete execution of the optimization pipeline from a fresh random seed, producing one final context. By "run-dependent" we meant that prompt updates from only the current batch are generation-local. Useful strategic lessons discovered in one round are not retained in the next, so independent end-to-end executions can drift toward different final contexts.
> > >
> > > On novelty, the right claim is not that each individual component is new in isolation. Prior lines of work treat these ingredients separately. MEMO turns inference-time context into a persistent, cross-generation object of optimization in multi-turn multi-agent games by coupling exploration with persistent memory. Memory is not just post-hoc reflection, and exploration is not just batch-local prompt editing. The two are linked across generations. The ablations in Table 3 in our manuscript confirm this coupling matters. Removing either the memory or exploration pathway substantially weakens the result.
> > >
> > > ---
> > >
> > > > **Q3**: ... on what basis do you claim "as sample size grows, it converges to the true win rate" ...
> > >
> > > You are right that our earlier sentence “as sample size grows, it converges to the true win rate” was imprecise, and we should not have stated it that way.
> > >
> > > Our intended point was narrower. In the **small-sample tournament regime**, raw empirical win rate can be noisy. TrueSkill is regularized by uncertainty, so early lucky contexts are less likely to be promoted than under raw win rate alone.
> > >
> > > The **value** of this choice is improved robustness to noisy finite-sample estimates.
> > >
> > > The corresponding **risk** is that strong but under-tested contexts may be temporarily under-ranked.
> > >
> > > Our mitigation is that strong candidates can persist and be re-evaluated in later generations rather than being decided by a single small batch. We will correct this and make the trade-off explicit.
> > >
> > > We ran fresh experiments comparing TrueSkill, ExpectedReturn, and WinRate as the ranking criterion on KuhnPoker across three runs.
> > >
> > > ExpectedReturn is defined as $\hat{J}(c) = \frac{W - L}{N}$, where $W$, $L$, and $N$ are wins, losses, and total games.
> > >
> > > | **Method** | **$TrueSkill$** | **$WinRate$** | **$ExpectedReturn$** |
> > > | ---------- | -------------- | -------------- | -------------- |
> > > | MEMO   | 56.7% (3.0%)   | 50.4% (3.6%)   | 55.1% (1.6%)   |
> > >
> > > TrueSkill and ExpectedReturn yield similar performance, suggesting the practical impact of this choice is small in our setting.
> > >
> > >
> > > ---
> > >
> > > > **Q2**: ... dependence on the horizon? ... finite number of possible outcomes or ... a further restriction?
> > >
> > > By “discrete outcomes” we mean episodic terminal feedback of the kind used in the current paper, such as win / draw / loss, where each completed interaction yields a comparable outcome signal for tournament selection. So this is not merely “any finite set of outcomes” in the abstract.
> > >
> > > What matters is that the task provides a reasonably well-defined episode-level signal that can be compared across candidate contexts. This is also where horizon matters. If the horizon is very short or the state space is tiny, there is less room for persistent memory to help because prompt optimization alone can already cover much of the strategy space. If the horizon is long but success depends mainly on implicit computation or latent-state tracking that is hard to externalize in text, text memories become a weaker substrate. RL can be more effective in such cases. MEMO tends to help most when interactions are long enough for early decisions to compound and when useful lessons can be expressed and reused in language. We will state this scope condition more sharply in the revision.

---

### Official Review · Reviewer_5wpg · 2026-03-23

**Soundness:** 3
**Presentation:** 3
**Significance:** 3
**Originality:** 3
**Overall Recommendation:** 4
**Confidence:** 3

**Summary:**

This paper addresses the issues of context optimization and evaluation stability in multi-round, multi-agent games involving large language models. The authors note that in such interactive game environments, model performance is highly sensitive to prompt and context selection; subtle differences can trigger strategy shifts and further affect relative comparisons between methods. To address this, the paper proposes MEMO, a self-play context optimization framework that does not require updating model weights, treating the context during inference as an optimizable entity composed of the instruction prompt and external memory. By integrating structured reflection, persistent memory, context evolution, and replay mechanisms, the method enhances performance and reduces variability across runs within a limited self-play budget. Experimental results demonstrate that MEMO achieves competitive performance across multiple text-based game environments and exhibits notable advantages in terms of stability.

**Compliance With Llm Reviewing Policy:**

Affirmed.

**Final Justification:**

Thank you for the detailed response which addresses my concerns. I am therefore maintaining my positive score.

**Key Questions For Authors:**

1. Since the experiments focus on text-based game environments, it is still unclear how well the proposed framework would transfer to other interactive multi-agent settings.
2. Are the current baselines fully aligned with the problem setting in this paper, and could the authors include more directly relevant multi-agent comparison methods?
3. How do the authors explain the performance differences between MEMO and RL across different game types? In addition, Briscola shows large performance gaps across methods; could the authors further clarify the reasons and the specific characteristics of this task?

**Limitations:**

A main limitation is that the empirical study is restricted to text-based game environments. While this matches the target setting of the paper, it leaves the broader external validity of the framework underexplored.

**Strengths And Weaknesses:**

**Strengths**：
1. The empirical results are reasonably convincing. The paper evaluates the method on multiple text-based game environments and reports metrics such as average win rate and ranking stability, which support its central claim of improving performance while mitigating evaluation variance.
2. The method is fairly well designed. It integrates persistent memory, structured experience extraction, candidate selection, and prioritized replay into a unified self-play context optimization framework, resulting in a clear and reasonably systematic overall pipeline.

**Weaknesses**：
1. The evaluation is mainly conducted on text-based game environments, so it remains unclear whether the method generalizes to broader multi-agent tasks.
2. Some of the compared baselines are primarily designed for single-agent settings or general prompt optimization, and thus are not fully aligned with the multi-round multi-agent setting considered here.
3. The paper shows that the method performs differently across game types, but the underlying reasons for these differences are not analyzed in sufficient depth

---

> ### Author Rebuttal · Authors · 2026-03-31
>
> We thank the reviewer for appreciating our method framework and positive overview. Below, we address the concerns and questions brought up by the reviewer.
>
>
>
> > **W1/Q1**: Does MEMO generalize to broader multi-agent tasks?
>
> We conducted a new experiment on the **Bargaining** task from MultiAgentBench [1], a mixed-motive negotiation benchmark, optimizing GPT-4o-mini with MEMO and evaluating against three held-out seller models.
>
> | Seller | Method | Deal Rate | Savings (%) | Buyer Score |
> |---|---|---|---|---|
> | Gemini-2.5-Flash-Lite | Baseline | 73% | 17.5 | 3.67 |
> | | CoT | 83% | 18.8 | 3.71 |
> | | **MEMO** | **90%** | **19.4** | **3.80** |
> | Qwen3-235B | Baseline | 80% | 23.1 | 3.50 |
> | | CoT | **87%** | 23.9 | **3.79** |
> | | **MEMO** | 80% | **26.7** | **3.79** |
> | Grok-4-Fast | Baseline | 30% | 22.9 | 3.45 |
> | | CoT | 10% | 27.2 | 3.39 |
> | | **MEMO** | **60%** | **34.7** | **3.51** |
>
> These results confirm that MEMO generalizes from two-player zero-sum games to mixed-motive negotiation.
>
>
> > **W2/Q2**: Some of the compared baselines are primarily designed for single-agent settings or general prompt optimization, and thus are not fully aligned with the multi-round multi-agent setting considered here. ... Could the authors include more directly relevant multi-agent comparison methods?
>
> To our knowledge, no existing prompt optimization method targets multi-round multi-agent settings directly. TextGrad, MIPRO, and GEPA were designed for single-agent tasks; adaptations are in Appendix G in our manuscript. The two RL baselines, UnstableBaseline and SPIRAL, are natively multi-agent. We also added two memory-augmented baselines, Reflexion and mem0, in our response to Reviewer aUo5 Q2. MEMO outperforms both (+16.2% over Reflexion, +17.8% over mem0).
>
>
> ---
>
> > **W3/Q3**:
> >
> > The paper shows that the method performs differently across game types, but the underlying reasons for these differences are not analyzed in sufficient depth. ... Briscola shows large performance gaps across methods; could the authors further clarify the reasons and the specific characteristics of this task?
>
> MEMO's improvement varies across games, driven by how well winning strategies can be expressed in natural language and how much headroom the baseline leaves. The table ranks all five games by absolute gain on GPT-4o-mini.
>
> | Game | Category | Baseline | MEMO | Δ |
> |---|---|---|---|---|
> | Briscola | Imperfect info | 0.3% | 42.7% | +42.4% |
> | SimpleNegotiation | Negotiation | 31.3% | 54.9% | +23.6% |
> | SimpleTak | Perfect info | 21.4% | 41.8% | +20.4% |
> | TwoDollar | Negotiation | 32.2% | 52.4% | +20.2% |
> | KuhnPoker | Imperfect info | 39.1% | 55.6% | +16.5% |
>
> **Briscola (+42.4%).** The baseline achieves only 0.3% because Briscola's scoring hierarchy is counterintuitive, e.g., a Three beats a King and is worth more points. The model defaults to wrong card-rank intuitions, and MEMO's CRUD operations progressively correct this through accumulated insights.
>
> **Negotiation games (+23.6% SimpleNegotiation, +20.2% TwoDollar).** Negotiation strategies are inherently linguistic, so they translate directly into memory entries. The baseline is not fundamentally wrong but generic, and MEMO adds conditional specificity.
>
> **KuhnPoker (+16.5%).** With only 3 cards and a single betting round, even the baseline reaches 39.1%, leaving less headroom. The compact state space means prompt optimization alone can find reasonable strategies.
>
> ---
>
> > How do the authors explain the performance differences between MEMO and RL across different game types?
>
> MEMO extracts insights from intermediate decisions across full trajectories. RL learns only from the final win/loss and updates model weights. We compare both on Qwen2.5-7B-Instruct (MEMO: 2,000 games; UnstableBaseline: 38,000 games, REINFORCE + LoRA).
>
> | Game | Category | MEMO | UnstableBaseline | Δ |
> |---|---|---|---|---|
> | SimpleNegotiation | Negotiation | **48.0%** | 41.1% | +6.9% |
> | TwoDollar | Negotiation | **48.4%** | 30.4% | +18.0% |
> | KuhnPoker | Imperfect info | **60.0%** | 52.7% | +7.3% |
> | Briscola | Imperfect info | 31.1% | **53.3%** | −22.2% |
> | SimpleTak | Perfect info | 34.0% | **47.3%** | −13.3% |
> | **Mean** | | 44.3% | 45.0% | |
> | **RSE** | | **6.1%** | 43.3% | |
>
> MEMO outperforms RL on games with rich intermediate interactions, where each turn carries a learnable lesson, e.g., offer-response patterns or bluffing sequences. RL only sees win/loss at the end and must assign credit across many turns, making learning noisy.
>
> RL outperforms MEMO on games where critical reasoning is implicit computation that resists verbalization. SimpleTak requires spatial path-finding, and Briscola requires card-counting across ~40 turns. RL modifies model weights directly, learning implicit evaluation functions. MEMO achieves 6.1% RSE compared to UnstableBaseline's 43.3%, with 19× fewer training games.

---

> > ### Author Rebuttal · Reviewer_5wpg · 2026-04-04
> >
> > Thank you for the detailed response which addresses my concerns. I am therefore maintaining my positive score.

---

> > > ### Author Response · Authors · 2026-04-07
> > >
> > > We sincerely thank the reviewer for the positive evaluation and for confirming that our responses have adequately addressed the concerns. We are grateful for the thoughtful questions throughout the review process, which helped us strengthen the paper with additional experiments on mixed-motive negotiation and more detailed baseline comparisons. We appreciate the reviewer's decision to maintain the positive score.

---

### Official Review · Reviewer_6Mj5 · 2026-03-26

**Soundness:** 3
**Presentation:** 3
**Significance:** 3
**Originality:** 2
**Overall Recommendation:** 4
**Confidence:** 3

**Summary:**

The paper provides a weight-free self-play framework that optimizes inference-time context for LLMs in multi-turn, two-player text-based games. MEMO has two coupled components. The first is a retention mechanism, which maintains a persistent memory bank of strategic insights extracted from trajectory reflections, updated through Add, Edit, and Remove operations across optimization generations. The second is an exploration mechanism, which performs tournament-style context evolution using TrueSkill-based selection and prioritized replay of rare trajectory prefixes. The framework is evaluated on five text-based games spanning negotiation (SimpleNegotiation, TwoDollar), imperfect information (KuhnPoker, Briscola), and perfect information (SimpleTak). The authors have found that using GPT-4o-mini and Qwen-2.5-7B-Instruct, MEMO achieves substantial win rate improvements with only 2,000 self-play games per task, while also reducing run-to-run variance compared to baselines and other prompt optimization methods.

**Compliance With Llm Reviewing Policy:**

Affirmed.

**Final Justification:**

The rebuttal addressed most of my concerns.

**Key Questions For Authors:**

- How does the size of the memory bank evolve over the five optimization generations? Does it stabilize or keep growing?
- The memory bank samples a subset of insights for context injection. Could the authors clarify how this subset is selected? Would the author expect more structured retrieval, such as RAG or agentic retrieval where the model actively decides what to retrieve based on the current game state, could further improve performance?

**Limitations:**

yes.

**Strengths And Weaknesses:**

### Strength:

1. This paper addresses a timely and important problem. Besides scaling pre-training data or RLVR, working on zero-sum games where the equilibrium may produce interesting and transferable skills is an encouraging direction to improve models capability. The cross-game transfer results in Table 4 partially support this. Skills do transfer across games, which is encouraging.
2. The paper also resolves a run-to-run variance issue in multi-turn LLM game evaluations, consistently achieving the lowest RSE across methods.
3. The paper is technically sound. The tasks span different kinds of games including negotiation, imperfect information, and perfect information, and the models evaluated cover both open-source and closed-source models. The ablation study in Table 3 is well-done, clearly showing memory and exploration are essential to the performance improvement. The prompt sensitivity analysis in Appendix A is also a compelling result.
4. The paper is clearly constructed, and the report of limitation are candid. The retention and exploration components are clearly introduced. The paper is also honest about where MEMO falls short, which gives the reader a calibrated picture.

### Weaknesses

1. The horizon of the games tested are not so long, with KuhnPoker at roughly 6 actions and Briscola at around 40. The paper motivates MEMO by appealing to compounding deviations in long-horizon interactions in the abstract, but it would be more convincing with a longer-horizon game.
2. The use of "self-play" could be more precise. The opponent is always the same vanilla model with a default prompt and never adapts, whereas in classical self-play (AlphaGo/AlphaZero, cited in Section 6.3) both sides co-evolve. This is closer to one-sided context optimization against a static opponent. Though the current evaluation against different stronger models already suggests generalization, it would be interesting to see results where the opponent also improves,
3. The "multi-agent" framing in the title is technically correct since two-player games are multi-agent, but all experiments are strictly two-player zero-sum. Whether MEMO extends to 3+ player settings where challenges like coalition formation and non-transitive dynamics arise remains an open question.
4. Related work could be more complete. ExpeL (Zhao et al., AAAI 2024) uses operationally analogous insight management (Add, Edit, Upvote, Downvote), and STRATEGIST (Light et al., ICLR 2025), in particular, also builds text-based strategy libraries through self-play in games. Other earlier work like Voyager (Wang et al., 2023) and Reflexion (Shinn et al., NeurIPS 2023) also pioneer the idea of generating and maintaining strategies or skill libraries. Articulating how MEMO differs from these would help position the contribution.

---

> ### Author Rebuttal · Authors · 2026-03-31
>
> We thank the reviewer for highlighting the importance of the problem setting.
>
>
> > **W1**: The horizon of the games tested are not so long ... it would be more convincing with a longer-horizon game.
>
> We increased SimpleTak from 4×4 to 10×10, raising average game length from ~12 to ~62 turns. We optimized GPT-4o-mini with 2,000 self-play games and evaluated against two held-out opponents over 50 games each.
>
> | Opponent | Baseline | MEMO | Δ |
> |---|---|---|---|
> | Gemini-2.5-Flash-Lite | 54.0% | 64.0% | +10.0% |
> | Grok-4-Fast-Non-Reasoning | 46.0% | 58.0% | +12.0% |
> | **Mean** | **50.0%** | **61.0%** | **+11.0%** |
>
> MEMO improves mean win rate by +11.0%, confirming that gains extend beyond short-horizon games.
>
> ---
>
>
> > **W2**: The use of "self-play" could be more precise ... the opponent never adapts, whereas in classical self-play both sides co-evolve. ... It would be interesting to see results where the opponent also improves.
>
> We will clarify our setting as "one-sided context optimization self-play." We also ran a round-robin variant on KuhnPoker with GPT-4o-mini. Across three trials, the round-robin setup yields win rates of 55.6%, 58.6%, and 50.3%, with mean 54.8% and RSE 4.4%. Our one-sided setup yields 54.0%, 57.3%, and 55.3%, with mean 55.5% and RSE 1.8%. The means are comparable, but round-robin exhibits notably higher variance due to the added uncertainty of a co-evolving opponent. We will discuss this trade-off in the revision.
>
> ---
>
> > **W3**: All experiments are strictly two-player zero-sum. Whether MEMO extends to 3+ player settings where coalition formation and non-transitive dynamics arise remains an open question.
>
> MEMO is player-count agnostic: its components assume no fixed player count. We validated this on ThreePlayerIPD-v0, a three-player iterated prisoner's dilemma with free-chat communication and coalition dynamics, optimizing GPT-4o-mini with 2,000 self-play games.
>
> | **Method** | **vs Gemini** | **vs Grok** | **Average** |
> |---|---:|---:|---:|
> | Baseline | 8.0% | 21.3% | 14.7% |
> | CoT | 10.0% | 23.3% | 17.8% |
> | **MEMO** | **25.3%** | **32.0%** | **28.7%** |
>
> MEMO nearly doubles the baseline win rate from 14.7% to 28.7%, confirming it extends to settings with coalition formation and non-transitive dynamics.
>
>
> ---
>
> > **W4**: Related work could be more complete. ExpeL, STRATEGIST, Voyager, and Reflexion ... also pioneer the idea of generating and maintaining strategies or skill libraries. Articulating how MEMO differs from these would help position the contribution.
>
> We will add explicit comparisons to ExpeL, STRATEGIST, Voyager, and Reflexion in the revision, clarifying how MEMO's coupled exploration–retention design differs.
>
> ---
>
> > **Q1**: How does the memory bank size evolve over generations? Does it stabilize or keep growing?
>
> We analyzed memory bank evolution on TwoDollar with GPT-4o-mini across three runs.
>
> | | Gen 1 | Gen 2 | Gen 3 | Gen 4 |
> |---|---|---|---|---|
> | **Insights: Mean** | 20.3 (478) | 27.0 (663) | 32.0 (793) | 38.7 (937) |
> | **State Abstracts: Mean** | 10.0 (2,975) | 19.0 (5,700) | 19.7 (6,019) | 25.7 (8,357) |
>
> *Format: Unit Count, Token Count.*
>
> The bank grows but does not explode, because CRUD operations consolidate, rewrite, and remove redundant entries rather than simply appending. Without CRUD, an append-only strategy would accumulate over 160 insights by Gen 4; with CRUD the count is only 38.7 entries, remaining below 1K tokens. The growth rate is slowing but has not fully stabilized. Investigating principled mechanisms for controlling memory bank growth is an interesting direction for future work.
>
> ---
>
> > **Q2**:
> >
> > ... Could the authors clarify how the subset of insights for context injection is selected?
>
> At each generation, MEMO randomly samples a subset of insights from the memory bank to seed each candidate prompt. Each candidate in the population receives a different subset, so the population covers a range of strategic directions. The tournament then selects the best-performing candidates via self-play.
>
> > Would structured retrieval (e.g., RAG or agentic retrieval based on game state) further improve performance?
>
> We implemented a state-aware retrieval variant: at each turn, a retrieval LLM selects relevant insights and injects tactical guidance into the agent's context. We used GPT-4o-mini for both retrieval and play, and evaluated on KuhnPoker with 2,000 self-play games against three held-out opponents.
>
> | **Method** | **Mean Win Rate** |
> |---|---|
> | Baseline | 39.1% |
> | MEMO | **55.6%** |
> | MEMO + Agentic Retrieval | 54.7% |
>
> Agentic retrieval slightly decreases performance from 55.6% to 54.7%. First, MEMO's evolution loop already distills the most useful insights into the selected prompt, making runtime retrieval redundant. Second, the retrieval model occasionally injects incorrect reasoning, e.g., treating the Queen as a "strong card" when in KuhnPoker it is middle-strength.
>
> ---

---

> > ### Author Rebuttal · Reviewer_6Mj5 · 2026-04-03
> >
> > Thanks for the responses. I suggest discussing the use of "self-play" as a current limitation and future work, and a more detailed treatment on related work such as STRATEGIST.

---

> > > ### Author Response · Authors · 2026-04-07
> > >
> > > We sincerely thank the reviewer for carefully reading our responses and for confirming that the concerns have been adequately addressed. We also appreciate the two constructive suggestions.
> > >
> > > > **Self-play as a limitation**: ... I suggest discussing the use of "self-play" as a current limitation and future work ...
> > >
> > > We agree that this is an important point. As discussed in our response to W2, our current "one-sided context optimization self-play" setting does not involve a co-evolving opponent, which differs from classical self-play. We will add an explicit discussion of this distinction as a current limitation in the revision, along with the round-robin variant as a promising direction for future work.
> > >
> > > ---
> > >
> > > > **Related work on STRATEGIST**: ... a more detailed treatment on related work such as STRATEGIST.
> > >
> > > We will expand our related work discussion to include a more detailed comparison with STRATEGIST, articulating how MEMO's coupled exploration–retention design differs from STRATEGIST's approach. We appreciate this suggestion and believe it will strengthen the positioning of our contribution.

---

### Decision · Program_Chairs · 2026-04-30

**Decision:**

Accept (regular)

**Comment:**

The paper proposes a weight-free self-play framework that optimizes inference-time context for LLMs in multi-turn, two-player text-based games. The paper is well-motivated and well-implemented in terms of the retention and exploration mechanism. Experimental study is well-performed (significant better results than extensive baselines), though I find the benchmarks are relatively simple and narrow, as pointed out  by some reviewers as well. In general, the paper makes a solid contribution to the corresponding research community. The rebuttal went well, and all reviewers gave positive scores on this paper. Overall, I recommend acceptance and suggest the authors improve the paper further based on the rebuttal.